# Idempotence and Perceptual Image Compression

**Tongda Xu**[1,2]**, Ziran Zhu**[1,3]**, Dailan He**[4,5]**, Yanghao Li**[1,2]**, Lina Guo**[4]**, Yuanyuan Wang**[4]
[1]Institute for AI Industry Research, Tsinghua University
[2]Department of Computer Science and Technology, Tsinghua University

**Zhe Wang**[1,2]**, Hongwei Qin**[4]**, Yan Wang**[1*] **& Ya-Qin Zhang**[1,2,6*]
[3]Institute of Software, Chinese Academy of Sciences, [4]SenseTime Research
[5]The Chinese University of Hong Kong, [6]School of Vehicle and Mobility, Tsinghua University
x.tongda@nyu.edu, wangyan@air.tsinghua.edu.cn

## Abstract

Idempotence is the stability of image codec to re-compression. At the first glance, it is unrelated to perceptual image compression. However, we find that theoretically: 1) Conditional generative model-based perceptual codec satisfies idempotence; 2) Unconditional generative model with idempotence constraint is equivalent to conditional generative codec. Based on this newfound equivalence, we propose a new paradigm of perceptual image codec by inverting unconditional generative model with idempotence constraints. Our codec is theoretically equivalent to conditional generative codec, and it does not require training new models. Instead, it only requires a pre-trained mean-square-error codec and unconditional generative model. Empirically, we show that our proposed approach outperforms state-of-the-art methods such as HiFiC (Mentzer et al., 2020) and ILLM (Muckley et al., 2023), in terms of Fréchet Inception Distance (FID). The source code is provided in https://github.com/tongdaxu/Idempotence-and-Perceptual-Image-Compression.

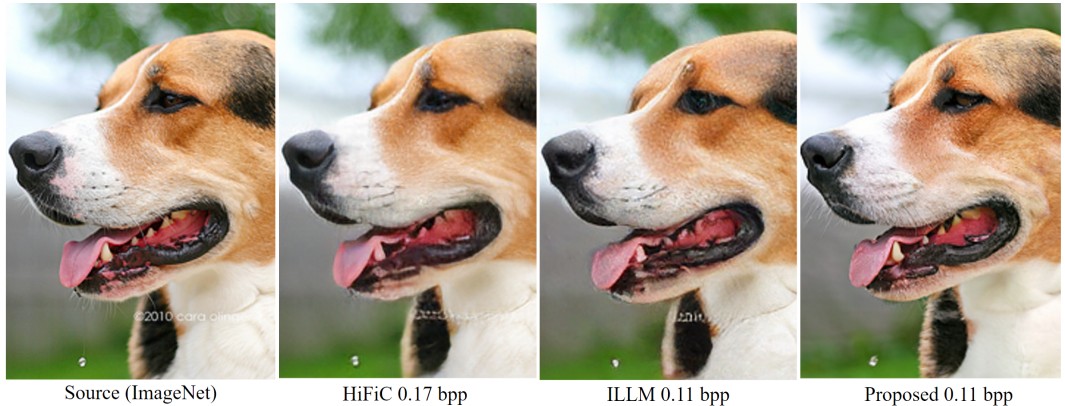

| Source (ImageNet) | HiFiC 0.17 bpp | ILLM 0.11 bpp | Proposed 0.11 bpp |

Figure 1: A visual comparison of our proposed approach with state-of-the-art perceptual image codec, such as HiFiC (Mentzer et al., 2020) and ILLM (Muckley et al., 2023).

## 1 Introduction

Idempotence refers to the stability of image code to re-compression, which is crucial to practical image codec. For traditional codec standard (e.g., JPEG (Wallace, 1991), JPEG2000 (Taubman et al.,

---

*To whom the correspondence should be addressed.

2002), JPEG-XL (Alakuijala et al., 2019)), idempotence is already taken into consideration. For neural image compression (NIC), by default idempotence is not considered. Thus, specific methods such as invertible network (Helminger et al., 2021; Cai et al., 2022; Li et al., 2023) and regularization loss (Kim et al., 2020) have been proposed to improve the idempotence of NIC methods.

In the meantime, there has been growing success in perceptual image compression. Many recent studies achieve perceptual near-lossless result with very low bitrate (Mentzer et al., 2020; Muckley et al., 2023). The majority of perceptual image compression methods adopt a conditional generative model (Tschannen et al., 2018; Mentzer et al., 2020; He et al., 2022b; Agustsson et al., 2022; Hoogeboom et al., 2023; Muckley et al., 2023). More specifically, they train a decoder that learns the posterior of a natural image conditioned on the bitstream. This conditional generative model-based approach is later theoretically justified by Blau & Michaeli (2019); Yan et al. (2021), who show that such approach achieves perfect perceptual quality and is optimal in terms of rate-distortion-perception trade-off when the encoder is deterministic.

At the first glance, idempotence and perceptual image compression are unrelated topics. Indeed, researchers in those two areas barely cite each other. However, we find that idempotence and perceptual image compression are, in fact, closely related. More specifically, we demonstrate that: 1) Perceptual image compression with conditional generative model brings idempotence; 2) Unconditional generative model with idempotence constraints brings perceptual image compression. Inspired by the latter, we propose a new paradigm of perceptual image compression, by inverting an unconditional generative model with idempotence constraint. Compared with previous conditional generative codec, our approach requires only a pre-trained unconditional generative model and mean-square-error (MSE) codec. Furthermore, extensive experiments empirically show that our proposed approach achieves state-of-the-art perceptual quality.

## 2 PRELIMINARIES

### 2.1 IDEMPOTENCE OF IMAGE CODEC

Idempotence of image codec refers to the codec's stability to re-compression. More specifically, denote the original image as $X$, the encoder as $f(.)$, the code as $Y = f(X)$, the decoder as $g(.)$ and the reconstruction as $\hat{X} = g(Y)$. We say that the codec is idempotent if

$$f(\hat{X}) = Y, \text{ or } g(f(\hat{X})) = \hat{X}, \tag{1}$$

i.e., the codec is idempotent if the re-compression of reconstruction $\hat{X}$ produces the same result.

### 2.2 PERCEPTUAL IMAGE COMPRESSION

In this paper, we use Blau & Michaeli (2018)'s definition of perceptual quality. More specifically, we say that the reconstructed image $\hat{X}$ has perfect perceptual quality if

$$p_{\hat{X}} = p_X, \tag{2}$$

where $p_X$ is the source image distribution and $p_{\hat{X}}$ is the reconstruction image distribution. In this paper, we slightly abuse the word "perception" for this divergence-based perception, and use the word "human perception" for human's perception instead.

The majority of perceptual image codec achieves perceptual quality by conditional generative model (Tschannen et al., 2018; Mentzer et al., 2020; He et al., 2022b; Agustsson et al., 2022; Hoogeboom et al., 2023). More specifically, they train a conditional generative model (such as conditional generative adversial network) to approximate the real image's posterior on the bitstream. And the reconstruction image $\hat{X}$ is obtained by sampling the posterior:

$$\hat{X} = g(Y) \sim p_{X|Y}, \text{ where } Y = f(X). \tag{3}$$

Blau & Michaeli (2019) prove that this conditional generative codec achieves perfect perceptual quality $p_{\hat{X}} = p_X$. Further, when $f(.)$ achieves the optimal MSE $\Delta^*$, then the MSE of perceptual reconstruction is bounded by twice of the optimal MSE:

$$\mathbb{E}[||X - \hat{X}||^2] \leq 2\Delta^*, \tag{4}$$

Moreover, Yan et al. (2021) further justify conditional generative codec by proving that it is optimal in terms of rate-distortion-perception (Blau & Michaeli, 2019) among deterministic encoders.

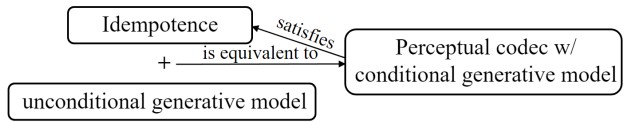

Figure 2: The relationship between idempotence and perceptual image compression.

## 3 IDEMPOTENCE AND PERCEPTUAL IMAGE COMPRESSION

In this section, we connect idempotence and perceptual image compression. Specifically, we show that ideal conditional generative codec is idempotent. On the other hand, sampling from unconditional generative model with ideal idempotence constraint leads to conditional generative codec. Their relationship is shown in Fig. 2.

### 3.1 PERCEPTUAL IMAGE COMPRESSION BRINGS IDEMPOTENCE

We first show that ideal conditional generative codec satisfies idempotence, i.e, $\hat{X} \sim P_{X|Y} \Rightarrow f(\hat{X}) = Y$. Given a specific value $y$ with non-trivial probability $p_Y(Y = y) \neq 0$, we define the inverse image of $y$ as $f^{-1}[y] = \{x|f(x) = y\}$. By definition, all the elements $x \in f^{-1}[y]$ encode into $y$. Then if we can show $\hat{X} \in f^{-1}[y]$, we can show that this perceptual codec is idempotent.

To show that $\hat{X} \in f^{-1}[y]$, let's consider a specific value $x$ with non-trivial probability $p_X(X = x) \neq 0$. As $Y = f(X)$ is a deterministic transform, when $x \notin f^{-1}[y]$, we have the likelihood $p_{Y|X}(Y = y|X = x) = 0$. Therefore, for any $x \notin f^{-1}[y]$, we have $p_{XY}(X = x, Y = y) = p_{Y|X}(Y = y|X = x)p_X(X = x) = 0$. As $p_{X|Y} = p_{XY}/p_Y$, for any $x \notin f^{-1}[y]$, we have the posterior $p_{X|Y}(X = x|Y = y) = 0$. And therefore, for any sample $\hat{X} \sim p_{X|Y=y}$, we have $\Pr\{\hat{X} \in f^{-1}[y]\} = 1$, i.e., $\hat{X} \in f^{-1}[y]$ almost surely. By definition of $f^{-1}[y]$, this codec is idempotent. We summarize the above discussion as follows:

**Theorem 1.** *(Perceptual quality brings idempotence) Denote $X$ as source, $f(.)$ as encoder, $Y = f(X)$ as bitstream, $g(.)$ as decoder and $\hat{X} = g(Y)$ as reconstruction. When encoder $f(.)$ is deterministic, then conditional generative model-based image codec is also idempotent, i.e.,*

$$\hat{X} = g(Y) \sim p_{X|Y} \Rightarrow f(\hat{X}) \overset{a.s.}{=} Y. \tag{5}$$

### 3.2 IDEMPOTENCE BRINGS PERCEPTUAL IMAGE COMPRESSION

In previous section, we have shown that perceptual quality brings idempotence. It is obvious that the simple converse is not true. A counter-example is JPEG2000 (Taubman et al., 2002), which is idempotent by design but optimized for MSE. However, we show that with unconditional generative model, idempotence does bring perceptual quality.

More specifically, we want to show that sampling from unconditional distribution $\hat{X} \sim p_X$ with idempotence constraint $f(\hat{X}) = Y$, is equivalent to sampling from the posterior $p_{X|Y}$. Again, we consider a non-trivial $y$ with $p_Y(Y = y) \neq 0$. Similar to previous section, as $f(.)$ is deterministic, we have $p_{Y|X}(Y = y|X = x) = 1$ if $x \in f^{-1}[y]$, and $p_{Y|X}(Y = y|X = x) = 0$ if $x \notin f^{-1}[y]$. Then, we can compute the posterior distribution as

$$p_{X|Y}(X = x|Y = y) \propto p_X(X = x)p_{Y|X}(Y = y|X = x)$$

$$\propto \begin{cases} p_X(X = x), & x \in f^{-1}[y], \\ 0, & x \notin f^{-1}[y]. \end{cases} \tag{6}$$

The above equation shows that when $x \notin f^{-1}[y]$, the posterior likelihood $p_{X|Y}(X = x|Y = y) = 0$, and no sample with value $x$ can be generated almost surely. And when $x \in f^{-1}[y]$, the posterior likelihood $p_{X|Y}(X = x|Y = y)$ is proportional to the source distribution $p_X(X = x)$. Therefore, sampling from the source $p_X$ with idempotence constraint $f(X) = Y$ is equivalent to sampling from the posterior $p_{X|Y}$. We summarize the above discussion as follows:

**Theorem 2.** *(Idempotence brings perceptual quality) Denote $X$ as source, $f(.)$ as encoder, $Y = f(X)$ as bitstream, $g(.)$ as decoder and $\hat{X} = g(Y)$ as reconstruction. When encoder $f(.)$ is deterministic, the unconditional generative model with idempotence constraint is equivalent to the conditional generative model-based image codec, i.e.,*

$$\hat{X} \sim p_X, \text{ s.t. } f(\hat{X}) = Y \Rightarrow \hat{X} \sim p_{X|Y}. \tag{7}$$

And more conveniently, the unconditional generative model with idempotence constraint also satisfies the theoretical results from rate-distortion-trade-off (Blau & Michaeli, 2019; Yan et al., 2021):

**Corollary 1.** *If $f(.)$ is the encoder of a codec with optimal MSE $\Delta^*$, then the unconditional generative model with idempotence constraint also satisfies*

$$p_{\hat{X}} = p_X, \mathbb{E}[||X - \hat{X}||^2] \leq 2\Delta^*. \tag{8}$$

*Furthermore, the codec induced by this approach is also optimal among deterministic encoders.*

Besides, those results can be extended to image restorations (See Appendix. A).

## 4 PERCEPTUAL IMAGE COMPRESSION BY INVERSION

### 4.1 GENERAL IDEA

Theorem 2 implies a new paradigm to achieve perceptual image compression by sampling from a pre-trained unconditional generative model with the idempotence constraint. More specifically, we can rewrite the left hand side of Eq. 7 in Theorem 2 as

$$\min ||f(\hat{X}) - Y||^2, \text{ s.t. } \hat{X} \sim p_X. \tag{9}$$

Then the problem has exactly the same form as a broad family of works named 'generative model inversion' for image super-resolution and other restoration tasks (Menon et al., 2020; Daras et al., 2021; Wang et al., 2022; Chung et al., 2022a). From their perspective, $f(.)$ is the down-sample operator, $Y$ is the down-sampled image and $||f(\hat{X}) - Y||^2$ is the 'consistency' penalization that secures the super-resolved image $\hat{X}$ corresponds to the input down-sampled image. The $\hat{X} \sim p_X$ is the 'realness' term, and it ensures that $\hat{X}$ lies on the natural image manifold (Zhu et al., 2016). And solving Eq. 9 is the same as finding a sample that satisfies the consistency, which is the inverse problem of sampling. Therefore, they name their approach 'inversion of generative model'. For us, the $f(.)$ operator is the encoder, and $Y$ is the bitstream. And so long as our encoder is differentiable, we can adopt their inversion approach for image super-resolution to inverse the codec.

### 4.2 ENCODE AND DECODE PROCEDURE

To better understand our proposed approach, we describe the detailed procedure of communicating an image from sender to receiver and implementation details. We start with the pre-trained model that is required for the sender and receiver:

- The sender and receiver share a MSE optimized codec with encoder $f_0(.)$, decoder $g_0(.)$. Despite we can use any base codec, MSE-optimized codec leads to tightest MSE bound.

- The receiver has a unconditional generative model $q_X$ approximating the source $p_X$.

And the full procedure of encoding and decoding an image with our codec is as follows:

- The sender samples an image from the source $X \sim p_X$.

- The sender encodes the image into bitstream $Y = f_0(X)$, with the encoder of pre-trained MSE codec. And $Y$ is transmitted from sender to receiver.

- Upon receiving $Y$, the receiver inverses a generative model with idempotence constraint:

$$\min ||f_0(\hat{X}) - Y||^2, \text{ s.t. } \hat{X} \sim q_X. \tag{10}$$

For practical implementation, we can not directly use the binary bitstream $Y$ for idempotence constraint. This is because most of generative model inversion methods (Menon et al., 2020; Daras et al., 2021; Chung et al., 2022a) use the gradient $\nabla_{\hat{X}} ||f_0(\hat{X}) - Y||^2$. On the other hand, for most NIC approaches, the entropy coding with arithmetic coding (Rissanen & Langdon, 1979) is not differentiable. We propose two alternative constraints to this problem:

- y-domain constraint: We do not constrain the actual bitstream $Y$. Instead, we constrain the quantized symbols before arithmetic coding. Further, we use straight-through-estimator to pass the gradient through quantization. And we call this constraint y-domain constraint.

- x-domain constraint: On the other hand, we can also decode the re-compressed image and constrain the difference between the MSE reconstructed image of source $X$ and the MSE reconstructed image of sample. More specifically, instead of solving Eq. 10, we solve:

$$\min ||g_0(f_0(\hat{X})) - g_0(Y)||^2, \text{ s.t. } \hat{X} \sim q_X, \tag{11}$$

Similarly, the quantization is relaxed by STE to allow gradient to pass. And we call this constraint x-domain constraint.

The y-domain and x-domain constraint correspond to two idempotence definitions in Eq. 1. And when $g_0(.)$ is MSE optimal, those two constraints are equivalent (See Theorem 3 of Appendix. A). And beyond STE, other gradient estimators such as additive noise, multi-sample noise and stochastic gumbel annealing can also be adopted (Ballé et al., 2017; Xu et al., 2022; Yang et al., 2020).

## 4.3 Comparison to Previous Work

Compared with previous works using conditional generative model (Mentzer et al., 2020; Muckley et al., 2023), our proposed approach does not require specific conditional generative model for different codec and bitrate. It only requires one unconditional generative model, and it can be directly applied on even pre-trained MSE codec. Furthermore, it is theoretically equivalent to conditional generative codec, which means that it conveniently shares the same theoretical properties in terms of rate-distortion-perception trade-off (Blau & Michaeli, 2019).

Compared with previous proof of concept using unconditional generative model (Ho et al., 2020; Theis et al., 2022), our proposed approach does not require bits-back coding (Townsend et al., 2018) or sample communication (Li & El Gamal, 2018), which might have rate or complexity overhead. Further, it is implemented as a practical codec. Moreover, our approach uses exactly the same bitstream $Y$ as MSE codec. This means that the receiver can choose to reconstruct a MSE optimized image or a perceptual image and even achieve perception-distortion trade-off (See Appendix. B.3).

## 5 Experiments

### 5.1 Experiment Setup

**Metrics** We use Fréchet Inception Distance (FID) to measure perceptual quality. This is because we use Blau & Michaeli (2018)'s definition of perceptual quality, which is the divergence between two image distributions. And FID is the most common choice for such purpose. We use MSE and Peak-Signal-Noise-Ratio (PSNR) to measure distortion, which are the default choice for image codec. We use bpp (bits per pixel) to measure bitrate. To compare codec with different bitrate, we adopt Bjontegaard (BD) metrics (Bjontegaard, 2001): BD-FID and BD-PSNR. Those BD-metrics can be seen as the average FID and PSNR difference between codec over their bitrate range.

**Datasets** Following previous works in unconditional image generation (Karras et al., 2019; Ho et al., 2020), we train our unconditional generative models on FFHQ (Karras et al., 2019) and ImageNet (Deng et al., 2009) dataset. As (Chung et al., 2022a), we split the first 1000 images of FFHQ as test set and the rest for training. And we use first 1000 images of ImageNet validation split as test set and use the ImageNet training split for training. To test the generalization ability of our method on other datasets, we also use first 1000 images of COCO (Lin et al., 2014) validation split and CLIC 2020 (Toderici et al., 2020) test split as additional test set. As previous works (Karras et al., 2019; Ho et al., 2020) in unconditional generative model, we central crop image by their short edge and rescale them to $256 \times 256$.

**Previous State-of-the-art Perceptual Codec** We compare our approach against previous state-of-the-art perceptual codec: HiFiC (Mentzer et al., 2020), Po-ELIC (He et al., 2022b), CDC (Yang & Mandt, 2023) and ILLM (Muckley et al., 2023). HiFiC is the first codec achieving perceptual near lossless compression with very low bitrate. Po-ELIC is the winner of CVPR CLIC 2022, a major competition for perceptual codec. CDC is a very recent perceptual codec using conditional diffusion model. ILLM is the latest state-of-the-art perceptual codec. We use the official model to test HiFiC, CDC and ILLM. We contact the authors of Po-ELIC to obtain the test results on our datasets. For fairness of comparison, we also test a re-trained version of HiFiC (Mentzer et al., 2020) and ILLM (Muckley et al., 2023) on FFHQ and ImageNet dataset. So that their training dataset becomes the same as ours. We acknowledge that there are other very competitive perceptual codec (Iwai et al., 2021; Ma et al., 2021; Agustsson et al., 2022; Goose et al., 2023; Hoogeboom et al., 2023), while we have not included them for comparison as they are either unpublished yet or do not provide pre-trained model for testing.

**MSE Codec Baseline** As we have discussed, our approach requires a MSE optimized codec as base model. As we only requires the codec to be differentiable, most of works in NIC (Ballé et al., 2018; Minnen et al., 2018; Minnen & Singh, 2020; Cheng et al., 2020; He et al., 2022a; Liu et al., 2023) can be used. Among those MSE codec, we choose two representative models: Hyper (Ballé et al., 2018) and ELIC (He et al., 2022a). Hyper is perhaps the most influential work in NIC. Its two level hyperprior structure inspires many later works. ELIC is the first NIC approach that outperforms state-of-the-art manual codec VTM (Bross et al., 2021) with practical speed. For reference, we also choose two traditional codec baseline: BPG and VTM (Bross et al., 2021).

|  | Base | Inversion | Constraint | bpp | FID ↓ | MSE ↓ | MSE bound |
|---|---|---|---|---|---|---|---|
| ELIC | - | - | - |  | 94.35 | 91.75 | - |
|  | StyleGAN2 | PULSE | x-domain |  | 15.33 | **754.2** |  |
|  | StyleGAN2 | ILO | x-domain | 0.07 | 26.15 | **689.2** |  |
| Proposed (ELIC) | DDPM | MCG | x-domain |  | 135.1 | **929.2** | 183.5 |
|  | DDPM | DPS | y-domain |  | 5.377 | 189.2 |  |
|  | DDPM | DPS | x-domain |  | **5.347** | 161.9 |  |

Table 1: Ablation study with FFHQ and ELIC. **Bold**: best FID. **Bold red**: too large MSE.

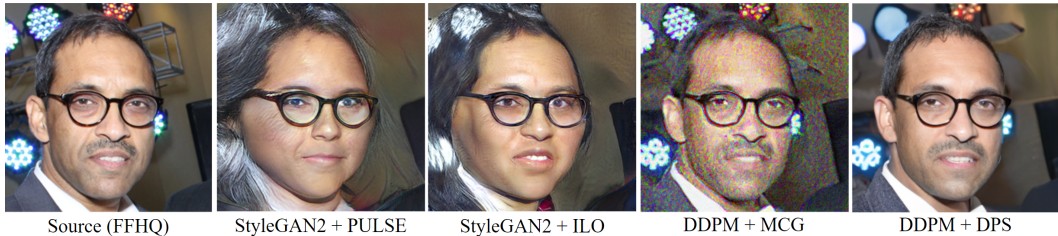

| Source (FFHQ) | StyleGAN2 + PULSE | StyleGAN2 + ILO | DDPM + MCG | DDPM + DPS |
|---|---|---|---|---|

Figure 3: Ablation study on unconditional generative model with FFHQ and ELIC.

## 5.2 ABLATION STUDY

**Unconditional Generative Model** As we have discussed, our approach requires an unconditional generative model for inversion. The most adopted models for inverse problem are StyleGAN family (Karras et al., 2019) and DDPM (Ho et al., 2020). Multiple approaches are proposed to invert StyleGAN and DDPM. And they are shown to be effective for image super-resolution and other restoration tasks (Menon et al., 2020; Daras et al., 2021; Roich et al., 2022; Daras et al., 2022; Wang et al., 2022; Chung et al., 2022b;a). For StyleGAN inversion, we choose two mostly adopted approach: PULSE (Menon et al., 2020) and ILO (Daras et al., 2021). For DDPM inversion, we choose MCG (Chung et al., 2022b) and DPS (Chung et al., 2022a) as they are the most popular methods for non-linear inversion.

To find the most suitable unconditional generative model, we compare StyleGAN2 + PULSE, Style-GAN2 + ILO, DDPM + MCG and DDPM + DPS. The implementation details are presented in Appendix. B.1. As Tab. 1 and Fig. 3 show, compared with DDPM + DPS, other three methods either have too high FID or have MSE significantly larger than 2×MSE of ELIC. Further, they look

visually less desirable and less similar to the source. Therefore, we use DDPM + DPS inversion in later experiments.

**Domain of Idempotence Constraint** As we have discussed, there are two kinds of constraint for idempotence, namely y-domain and x-domain. In theory, those two constraints are equivalent (See Theorem 3 of Appendix A). While in practice, they can be different as the strict idempotence condition $f_0(\hat{X}) = Y$ might not be achieved. We compare the y-domain and x-domain in Tab. 1. Those two constraints achieve similar FID while x-domain has lower MSE. This might be due to that x-domain directly optimizes MSE on pixel-level. Therefore, we choose x-domain constraint in later experiments.

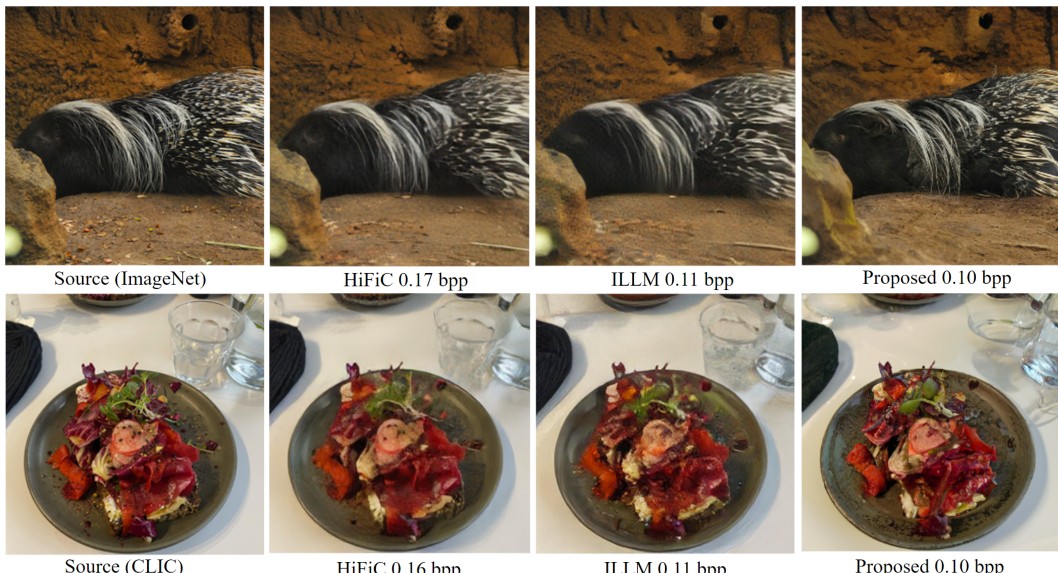

| Source (ImageNet) | HiFiC 0.17 bpp | ILLM 0.11 bpp | Proposed 0.10 bpp |

| Source (CLIC) | HiFiC 0.16 bpp | ILLM 0.11 bpp | Proposed 0.10 bpp |

Figure 4: A visual comparison of our proposed approach with state-of-the-art perceptual image codec, such as HiFiC (Mentzer et al., 2020) and ILLM (Muckley et al., 2023).

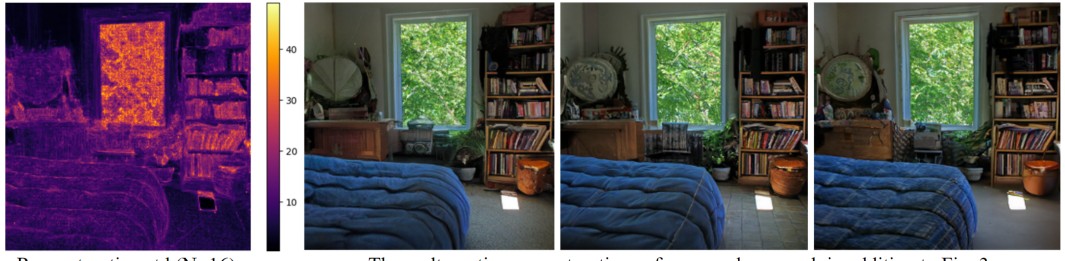

Reconstruction std (N=16)          Three alternative reconstructions of proposed approach in addition to Fig. 3.

Figure 5: Reconstruction diversity of proposed approach.

## 5.3 MAIN RESULTS

**Perceptual Quality** We compare the perceptual quality, in terms of FID, with state-of-the-art perceptual codec on multiple datasets. The results are shown in Tab. 2 and Fig. 11 of Appendix. B.3. Tab. 2 shows that our approach with ELIC (He et al., 2022a) achieves the lowest FID on all datasets. Furthermore, our approach with Hyper (Ballé et al., 2018) achieves second lowest FID on all datasets. We note that the base codec of HiFiC and ILLM is Mean-Scale Hyper (Minnen et al., 2018), which outperforms the Hyper. On the other hand, the base codec of Po-ELIC is ELIC, which is the same as ELIC. Besides, our approach outperforms CDC, which uses DDPM as generative model like us. Additionally, on FFHQ and ImageNet dataset, our approach also outperforms HiFiC and ILLM re-trained on those two dataset. Therefore, it is clear that our approach outperforms previous perceptual codec and achieves state-of-the-art FID metric, as we have excluded the difference

of base codec, generative model and dataset. Furthermore, qualitative results in Fig. 1, Fig. 4 and Fig. 12- 15 of Appendix B.3 also show that our approach is visually more desirable. On the other hand, Fig. 11 of Appendix B.3 shows that the MSE of our approach is within twice of base codec, and Tab. 2 shows that the PSNR of our method is within $10 \log_{10} 2 = 3.01$ dB of base codec. This means that our proposed approach satisfies the MSE bound by rate-distortion-perception trade-off (Blau & Michaeli, 2018). And as our approach shares bitstream with MSE codec, we can achieve perception-distortion trade-off (See Appendix. B.3). In addition, we also evaluate KID (Binkowski et al., 2018) and LPIPS (Zhang et al., 2018) in Appendix. B.3.

| Method | FFHQ | | ImageNet | | COCO | | CLIC | |
|---|---|---|---|---|---|---|---|---|
| | BD-FID ↓ | BD-PSNR ↑ | BD-FID ↓ | BD-PSNR ↑ | BD-FID ↓ | BD-PSNR ↑ | BD-FID ↓ | BD-PSNR ↑ |
| *MSE Baselines* | | | | | | | | |
| Hyper | 0.000 | 0.000 | 0.000 | 0.000 | 0.000 | 0.000 | 0.000 | 0.000 |
| ELIC | -9.740 | 1.736 | -10.50 | 1.434 | -8.070 | 1.535 | -10.23 | 1.660 |
| BPG | -4.830 | -0.8491 | -8.830 | -0.3562 | -4.770 | -0.3557 | -4.460 | -0.4860 |
| VTM | -14.22 | 0.7495 | -13.11 | 0.9018 | -11.22 | 0.9724 | -12.21 | 1.037 |
| *Conditional Generative Model-based* | | | | | | | | |
| HiFiC | -48.35 | -2.036 | -44.52 | -1.418 | -44.88 | -1.276 | -36.16 | -1.621 |
| HiFiC* | -51.85 | -1.920 | -47.18 | -1.121 | - | - | - | - |
| Po-ELIC | -50.77 | 0.1599 | -48.84 | 0.1202 | -50.81 | 0.2040 | -42.96 | 0.3305 |
| CDC | -43.80 | -8.014 | -41.75 | -6.416 | -45.35 | -6.512 | -38.31 | -7.043 |
| ILLM | -50.58 | -1.234 | -48.22 | -0.4802 | -50.67 | -0.5468 | -42.95 | -0.5956 |
| ILLM* | -52.32 | -1.415 | -47.99 | -0.7513 | - | - | - | - |
| *Unconditional Generative Model-based* | | | | | | | | |
| Proposed (Hyper) | -54.14 | -2.225 | -52.12 | -2.648 | -56.70 | -2.496 | -44.52 | -2.920 |
| Proposed (ELIC) | **-54.89** | -0.9855 | **-55.18** | -1.492 | **-58.45** | -1.370 | **-46.52** | -1.635 |

Table 2: Results on FFHQ, ImageNet, COCO and CLIC. *: re-trained on corresponding dataset. **Bold**: lowest FID. Underline: second lowest FID.

**Diversity of Reconstruction** Another feature of our approach is the reconstruction diversity. Though it is theoretically beneficial to adopt stochastic decoder for conditional generative codec (Freirich et al., 2021), most of previous works (Mentzer et al., 2020; He et al., 2022b; Muckley et al., 2023) adopt deterministic decoder and lost reconstruction diversity. On the other hand, our approach preserves this diversity. In Fig. 5, we show the pixel-level standard deviation $\sigma$ of our reconstruction on the first image of Fig. 4 with sample size 16. Further, in Fig. 5, 16 of Appendix B.3, we present alternative reconstructions, which differ a lot in detail but all have good visual quality.

**Idempotence** Consider we have a MSE codec. The first time compression is $\hat{X}^{(1)} = g_0(f_0(X))$, and re-compression is $\hat{X}^{(2)} = g_0(f_0(\hat{X}^{(1)}))$. We evaluate its idempotence by MSE between first time compression and re-compression $||\hat{X}^{(1)} - \hat{X}^{(2)}||^2$. According to our theoretical results, we can also use our approach to improve idempotence of base codec. Specifically, after first time compression, we can decode a perceptual reconstruction $\hat{X}_p^{(1)}$ by Eq. 10, and use this perceptual reconstruction for re-compression $\hat{X}^{(2)'} = g_0(f_0(\hat{X}_p^{(1)}))$. By Theorem 1, we should have $||\hat{X}^{(1)} - \hat{X}^{(2)'}||^2 = 0$, i.e., this augmented codec is strictly idempotent. In practice, as shown in Tab. 3, the augmented codec's re-compression MSE is smaller than the base codec. This indicates that our proposed approach can acts as an idempotence improvement module for MSE codec.

| | Re-compression metrics | |
|---|---|---|
| | MSE ↓ | PSNR (dB) ↑ |
| Hyper | 6.321 | 40.42 |
| Hyper w/ Proposed | 2.850 | 44.84 |
| ELIC | 11.80 | 37.60 |
| ELIC w/ Proposed | 7.367 | 40.93 |

Table 3: The idempotence comparison between base MSE codec and our approach.

| Method | Number of models | Train | Test |
|---|---|---|---|
| HiFiC, ILLM | $K$ | $\sim K$ weeks | ~0.1s |
| MCG, DPS | 1 | 0 | ~50s |
| Proposed | 1 | 0 | ~60s |

Table 4: Complexity of different methods, $K$ refers to the number of bitrate supported, which is 3 for HiFiC and 6 for ILLM.

**Complexity** Compared with conditional generative codec (e.g., HiFiC (Mentzer et al., 2020) and ILLM (Muckley et al., 2023)), our approach has lower training but higher testing complexity. Compared with inversion based image super-resolution (e.g. MCG (Chung et al., 2022b), DPS (Chung et al., 2022a)), our approach's complexity is similar. Tab. 4 shows that our approach greatly reduces training time. Specifically, for conditional generative codec, a conditional generative model is required for each rate. If the codec supports $K$ rates, $K$ generative models should be trained. For HiFiC and ILLM, each model takes approximately 1 week. And the codec needs $K$ weeks of training. While for us, this extra training time is 0 as we can utilize pre-trained unconditional generative model. While as the inversion-based image super-resolution, our method requires gradient ascent during testing, which increases the testing time from $\sim 0.1$s to $\sim 60$s.

## 6    RELATED WORK

### 6.1    IDEMPOTENT IMAGE COMPRESSION

Idempotence is a consideration in practical image codec (Joshi et al., 2000). In traditional codecs like JPEG (Wallace, 1991) and JPEG2000 (Taubman et al., 2002), the encoding and decoding transforms can be easily made invertible, thus idempotence can be painlessly achieved in these codec. In NIC, however, it requires much more efforts to ensure the idempotence (Kim et al., 2020). This is because neural network based transforms are widely adopted in NIC, and making these transforms invertible either hinders the RD performance (Helminger et al., 2021) or complicates the coding process (Cai et al., 2022). To the best of our knowledge, we are the first to build theoretical connection between idempotence and perceptual quality.

### 6.2    PERCEPTUAL IMAGE COMPRESSION

The majority of perceptual image codec adopt conditional generative models and successfully achieve near lossless perceptual quality with very low bitrate (Rippel & Bourdev, 2017; Tschannen et al., 2018; Mentzer et al., 2020; Agustsson et al., 2022; Yang & Mandt, 2023; Muckley et al., 2023; Hoogeboom et al., 2023). Blau & Michaeli (2019) show that conditional generative codec achieves perfect perceptual quality with at most twice of optimal MSE. Yan et al. (2021) further prove that this approach is the optimal among deterministic encoders. The unconditional generative model-based codec is explored by Ho et al. (2020); Theis et al. (2022). Though no actual codec is implemented, they reveal the potential of unconditional generative model in image compression. To the best of our knowledge, our approach is the first actual codec using unconditional generative model. It does not require training new models and is equivalent to conditional generative model-based perceptual codec.

## 7    DISCUSSION & CONCLUSION

A major limitation of our proposed approach is the testing time. This limitation is shared by all methods that use inversion of generative models (Menon et al., 2020; Daras et al., 2021; Chung et al., 2022a), and there are pioneering works trying to accelerate it (Dinh et al., 2022). Another limitation is that the resolution of the proposed approach is not as flexible. We can use patches as workaround (Hoogeboom et al., 2023) but there can be consistency issue. This limitation is also shared by all unconditional generative models, and there are also pioneering works trying to solve it (Zhang et al., 2022). As those two limitations are also important in broader generative modeling community, we believe they will be solved soon as the early-stage methods grow mature.

To conclude, we reveal that idempodence and perceptual image compression are closely connected. We theoretically prove that conditional generative codec satisfies idempotence, and unconditional generative model with idempotence constrain is equivalent to conditional generative codec. Based on that, we propose a new paradigm of perceptual codec by inverting unconditional generative model with idempotence constraint. Our approach does not require training new models, and it outperforms previous state-of-the-art perceptual codec.

ETHICS STATEMENT

Improving the perceptual quality of NIC in low bitrate has positive social values, including reducing carbon emission by saving resources for image transmission and storage. However, there can also be negative impacts. For example, in FFHQ dataset, a face with identity different from the original image can be reconstructed. And this mis-representation problem can bring issue in trustworthiness.

REPRODUCIBILITY STATEMENT

For theoretical results, the proof for all theorems are presented in Appendix A. For experiment, all four datasets used are publicly accessible. In Appendix B.1, we provide additional implementation details including the testing scripts for baselines and how we tune hyper-parameters. Besides, we provide source code for reproducing the experimental results as supplementary material.

ACKNOWLEDGMENTS

Funded by National Science and Technology Major Project (2022ZD0115502) and Baidu Inc. through Apollo-AIR Joint Research Center.

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

## A  PROOF OF MAIN RESULTS

In previous sections we give intuition and proof sketch of the theoretical results. We provide formal proof in this appendix.

**Notations** We use the capital letter $X$ to represent discrete random variable, lowercase letter $x$ to represent a specific realization of random variable, and calligraphic letter $\mathcal{X}$ to represent the alphabet of a random variable. We use $p_X$ to represent the probability law of random variable and $p_X(X = x)$ to represent the probability of a specific realization of a discrete random variable. We use $\Pr(.)$ to represent the probability of a specific event. We use $\overset{\Delta}{=}$ to represent definition and $\overset{a.s.}{=}$ to represent almost sure convergence.

**Theorem 1.**  *(Perceptual quality brings idempotence) Denote $X$ as source, $f(.)$ as encoder, $Y = f(X)$ as bitstream, $g(.)$ as decoder and $\hat{X} = g(Y)$ as reconstruction. When encoder $f(.)$ is deterministic, then conditional generative model-based image codec is also idempotent, i.e.,*

$$\hat{X} = g(Y) \sim p_{X|Y} \Rightarrow f(\hat{X}) \overset{a.s.}{=} Y.$$

*Proof.* We start with a specific value $y \in \mathcal{Y}$, where $\mathcal{Y}$ is the alphabet of random variable $Y$. Without loss of generality, we assume $p_Y(Y = y) \neq 0$, i.e., $y$ lies within the support of $p_Y$. We define the inverse image of $y$ as a set:

$$f^{-1}[y] \overset{\Delta}{=} \{x \in \mathcal{X} | f(x) = y\}, \tag{12}$$

where $\mathcal{X}$ is the alphabet of random variable $X$. According to the definition of idempotence, we need to show that $\hat{X} \in f^{-1}[y]$. Note that as encoder $f(.)$ is deterministic, each $x \in \mathcal{X}$ only corresponds to one $y$. Again, we consider a specific value $x \in \mathcal{X}, p_X(X = x) \neq 0$. We note that the likelihood of $Y$ can be written as

$$p_{Y|X}(Y = y | X = x) = \begin{cases} 1, & f(x) = y \\ 0, & f(x) \neq y \end{cases} \tag{13}$$

Then, for all $x \notin f^{-1}[y]$, the joint distribution $p_{XY}(X = x, Y = y) = p_X(X = x)p_{Y|X}(Y = y | X = x) = 0$. And thus the posterior $p_{X|Y}(X = x, Y = y) = 0$. In other words, for all samples $\hat{X} \sim p_{X|Y}(X | Y = y)$, the event $\Pr(\hat{X} \notin f^{-1}[y]) = 0$. And therefore, we conclude that

$$\Pr(\hat{X} \in f^{-1}[y]) = 1, \tag{14}$$

which indicates almost sure convergence $f(\hat{X}) \overset{a.s.}{=} Y$.  $\square$

**Theorem 2.**  *(Idempotence brings perceptual quality) Denote $X$ as source, $f(.)$ as encoder, $Y = f(X)$ as bitstream, $g(.)$ as decoder and $\hat{X} = g(Y)$ as reconstruction. When encoder $f(.)$ is deterministic, the unconditional generative model with idempotence constraint is equivalent to the conditional generative model-based image codec, i.e.,*

$$\hat{X} \sim p_X, \text{ s.t. } f(\hat{X}) = Y \Rightarrow \hat{X} \sim p_{X|Y}.$$

*Proof.* Similar to the proof of Theorem 1, we consider a specific value of $x \in \mathcal{X}$ with $p_X(X = x) \neq 0$. As

$$y = f(x) \tag{15}$$

is a deterministic transform, we have

$$p_{Y|X}(Y = y | X = x) = \begin{cases} 1, & f(x) = y \\ 0, & f(x) \neq y \end{cases} \tag{16}$$

Then by Bayesian rule, for each $(x, y) \in \mathcal{X} \times \mathcal{Y}$,

$$p_{X|Y}(X = x | Y = y) \propto p_{Y|X}(Y = y | X = x)p_X(X = x) \tag{17}$$

$$\propto h(X = x, Y = y), \tag{18}$$

$$\text{where } h(X = x, Y = y) = \begin{cases} 1 \times p_X(X = x) = p_X(X = x), & f(x) = y \\ 0 \times p_X(X = x) = 0, & f(x) \neq y \end{cases} \tag{19}$$

We can treat $h(X = x, Y = y)$ as an un-normalized joint distribution. $\forall (x, y) \in \mathcal{X} \times \mathcal{Y}$, it has 0 mass where $f(x) \neq y$, and a mass proportional to $p_X(X = x)$ where $f(x) = y$. As this holds true $\forall (x, y)$, then we have

$$p_{X|Y} \propto h(X, Y) \tag{20}$$

And sampling from this un-normalized distribution is equivalent to sampling from $p_X$ with the constrain $f(X) = y$. Therefore, sampling from posterior

$$\hat{X} \sim p_{X|Y}(X|Y) \tag{21}$$

is equivalent to the constrained sampling from marginal

$$\hat{X} \sim p_X, \text{ s.t. } f(\hat{X}) = Y, \tag{22}$$

which completes the proof. $\square$

**Corollary 1.** *If $f(.)$ is the encoder of a codec with optimal MSE $\Delta^*$, then the unconditional generative model with idempotence constraint also satisfies*

$$p_{\hat{X}} = p_X, \mathbb{E}[||X - \hat{X}||^2] \leq 2\Delta^*. \tag{23}$$

*Furthermore, the codec induced by this approach is also optimal among deterministic encoders.*

*Proof.* By Theorem 2, we have shown the equivalence of the unconditional generative model with idempotence constraint and conditional generative model. Then this corollary is simply applying Theorem 2 of Blau & Michaeli (2019) and Theorem 2, Theorem 3 of Yan et al. (2021) to conditional generative model. $\square$

In previous section, we state that for MSE optimal codec, the idempotence constraint $f(\hat{X}) = Y$ is equivalent to $g(f(\hat{X})) = g(Y)$. Now we prove it in this appendix. To prove this, we only need to show that for MSE optimal codec, the decoder $g(.)$ is a invertible mapping. Blau & Michaeli (2019) already show that $g(.)$ is a deterministic mapping, i.e., $\forall y_1, y_2 \in \mathcal{Y}, g(Y_1) \neq g(Y_2) \Rightarrow y_1 \neq y_2$. Then, we only need to show that if the reconstruction is different, the bitstream is different, i.e., $\forall y_1, y_2 \in \mathcal{Y}, y_1 \neq y_2 \Rightarrow g(Y_1) \neq g(Y_2)$. Formally, we have:

**Theorem 3.** *Denote $X$ as source, $f(.)$ as encoder, $Y = f(X)$ as bitstream, $g(.)$ as decoder and $\hat{X} = g(Y)$ as reconstruction. When encoder $f(.)$ is deterministic, for MSE optimal codec,*

$$y_1 \neq y_2 \Rightarrow g(y_1) \neq g(y_2). \tag{24}$$

*Proof.* We prove by contradiction. We assume that $\exists y_1 \neq y_2, g(y_1) = g(y_2)$. Then we notice that we can always construct a new codec, with bitstream $y_1, y_2$ merged into a new one with higher probability $p_Y(Y = y_1) + p_Y(Y = y_2)$. This means that this new codec has exactly the same reconstruction, while the bitrate is lower. This is in contradiction to the assumption that our codec is MSE optimal. $\square$

To provide a more intuitive illustration, we will include an example with discrete finite alphabet (our theoretical results are applicable to any countable alphabet or Borel-measurable source $X$ and any deterministic measureable function $f(.)$):

**Example 1.** *(1-dimension, discrete finite alphabet, optimal 1-bit codec): Consider discrete 1d random variable $X$, with alphabet $\mathcal{X} = \{0, 1, 2, 3, 4, 5\}$, and $Y$ with alphabet $\mathcal{Y} = \{0, 1\}$. We assume $P(X = i) = \frac{1}{6}$, i.e., $X$ follows uniform distribution. We consider a deterministic transform $f(X) : \mathcal{X} \to \mathcal{Y} = round(X/3)$. Or to say, $f(.)$ maps $\{0, 1, 2\}$ to $\{0\}$, and $\{3, 4, 5\}$ to $\{1\}$. This is infact the MSE-optimal 1-bit codec for source $X$ (See Chapter 10 of [Elements of Information Theory]). And the joint distribution $P(X, Y)$, posterior $P(X|Y)$ is just a tabular in Tab. 5.*

*We first examine Theorem. 1, which says that any sample from $X \sim P(X|Y = y)$ satisfies $f(X) = y$. Observing this table, this is indeed true. As for $f(X) \neq y$, the posterior $P(X = x|Y = y)$ is 0.*

*We then examine Theorem. 2, which says that sampling from $X \sim P(X)$ with $f(X) = y$ constraint is the same as sampling from $P(X|Y = y)$. We first identify the set such that $f(X) = y$. When $y = 0$, this set is $\{0, 1, 2\}$. And when $y = 1$, this set is $\{3, 4, 5\}$. And sampling from $p(X)$ with*

| x | y | $p(X = x, Y = y)$ | $p(X = x \mid Y = y)$ | $f(x)$ |
|---|---|---|---|---|
| 0 | 0 | 1/6 | 1 | 0 |
| 1 | 0 | 1/6 | 1 | 0 |
| 2 | 0 | 1/6 | 1 | 0 |
| 3 | 0 | 0 | 0 | 1 |
| 4 | 0 | 0 | 0 | 1 |
| 5 | 0 | 0 | 0 | 1 |
| 0 | 0 | 0 | 0 | 0 |
| 1 | 1 | 0 | 0 | 0 |
| 2 | 1 | 0 | 0 | 0 |
| 3 | 1 | 1/6 | 1 | 1 |
| 4 | 1 | 1/6 | 1 | 1 |
| 5 | 1 | 1/6 | 1 | 1 |

Table 5: Tabular of probability in Example. 1.

*constrain $f(X) = y$ is equivalent to sampling from one of the two subset with probability $\propto p(X)$.
And this probability is exactly $P(X \mid Y = y)$.*

Additionally, Note that we only limit $f(.)$ to be deterministic and measureable. Thus, those theoretical results can be extended to other noise-free inversion-based image restoration, such as super-resolution. Despite inversion-based super-resolution has been studied for years empirically, their theoretical relationship with conditional model based super-resolution, and distortion-perception (Blau & Michaeli, 2018) trade-off is in general unknown. PULSE (Menon et al., 2020) are the pioneer of this area, and they justify their approach by "natural image manifold" assumption. And later works in inversion-based super-resolution follow theirs story. On the other hand, our Theorem. 1, 2 and Corollary. 3 can be extended into image super-resolution as:

- Conditional generative super-resolution also satisfies idempotence, that is, the up-sampled image can down-sample into low-resolution image.
- Inversion-based super-resolution is theoretically equvalient to conditional generative super-resolution.
- Inversion-based super-resolution satisfies the theoretical results of distortion-perception trade-off (Blau & Michaeli, 2018), that is the MSE is at most double of best MSE.

We believe that our result provides non-trivial theoretical insights to inversion-based super-resolution community. For example, Menon et al. (2020) claim that the advantage of inversion-based super-resolution over the conditional generative super-resolution is that inversion-based super-resolution "downscale correctly". However, with our theoretical result, we know that ideal conditional generative super-resolution also "downscale correctly". Currently they fail to achieve this due to implementing issue. Another example is that most inversion-based super-resolution (Menon et al., 2020) (Daras et al., 2021) report no MSE comparison with sota MSE super-resolution, as it is for sure that their MSE is worse and there seems to be no relationship between their MSE and sota MSE. However, with our theoretical result, we know that their MSE should be smaller than 2x sota MSE. And they should examine whether their MSE falls below 2x sota MSE.

After the submission deadline, we become aware of an alternative of Theorem. 1 and Theorem. 2 which is presented in a concurrent paper by Ohayon et al. (2023).

## B  ADDITIONAL EXPERIMENTS

### B.1  ADDITIONAL EXPERIMENT SETUP

All the experiments are implemented in Pytorch, and run in a computer with AMD EPYC 7742 CPU and Nvidia A100 GPU.

For FID evaluation, we adopt the same code as official implementation of OASIS (Sushko et al., 2020) in `https://github.com/boschresearch/OASIS`. To ensure we have enough num-

ber of sample for FID, we slice the images into $64 \times 64$ non-overlapping patches. All the BD metrics are computed over bpp $0.15 - 0.45$, this is because HiFiC and CDC only have the bitrate over this range. And extrapolation beyond this range can cause inaccurate estimation.

For the re-trained version of HiFiC (Mentzer et al., 2020), we adopt a widely used Pytorch implementation in `https://github.com/Justin-Tan/high-fidelity-generative-compression`. For the re-trained version of ILLM (Muckley et al., 2023), we adopt the official Pytorch implementation. We follow the original paper to train our models on FFHQ and ImageNet dataset.

We note that the official HiFiC (Mentzer et al., 2020) is implemented in TensorFlow in `https://github.com/tensorflow/compression`. We test the official implementation and the Pytorch implementation, and the results are as Tab. 6. In terms of BD-FID, the Pytorch version outperforms Tensorflow version in FFHQ, ImageNet and CLIC dataset, while it is outperformed by tf version in COCO. The overall conclusion is not impacted.

| Method | FFHQ | | ImageNet | | COCO | | CLIC | |
|---|---|---|---|---|---|---|---|---|
| | BD-FID ↓ | BD-PSNR ↑ | BD-FID ↓ | BD-PSNR ↑ | BD-FID ↓ | BD-PSNR ↑ | BD-FID ↓ | BD-PSNR ↑ |
| HiFiC (Pytorch) | -48.35 | -2.036 | -44.52 | -1.418 | -44.88 | -1.276 | -36.16 | -1.621 |
| HiFiC (Tensorflow) | -46.44 | -1.195 | -40.25 | -0.828 | -46.45 | -0.917 | -35.90 | -0.920 |

Table 6: Comparison of Pytorch HiFiC and Tensorflow HiFiC.

For Hyper (Ballé et al., 2018), we use the pre-trained model by CompressAI (Bégaint et al., 2020), which is trained on Vimeo (Xue et al., 2017). For ELIC (He et al., 2022a), we use the pre-trained model in `https://github.com/VincentChandelier/ELiC-ReImplemetation`, which is trained on ImageNet training split. To the best of our knowledge, their training dataset has no overlap with our test set.

For BGP, we use the latest version BPG 0.9.8 in `https://bellard.org/bpg/`. For VTM, we use the latest version VTM 22.0 in `https://vcgit.hhi.fraunhofer.de/jvet/VVCSoftware_VTM/-/releases/VTM-22.0`. We convert the source to YUV444, encode with VTM and convert back to RGB to compute the metrics. The detailed command line for BPG and VTM are as follows:

```
bpgenc -q {22,27,32,37,42,47} file -o filename.bpg

bpgdec filename.bpg -o file

./EncoderApp -c encoder_intra_vtm.cfg -i filename.yuv \
-q {22,27,32,37,42,47} -o /dev/null -b filename.bin \
--SourceWidth=256 --SourceHeight=256 --FrameRate=1 \
--FramesToBeEncoded=1 --InputBitDepth=8 \
--InputChromaFormat=444 --ConformanceWindowMode=1

./DecoderApp -b filename.bin -o filename.yuv -d 8
```

For training of StyleGAN (Karras et al., 2019), we adopt the official implementation. The only difference is that our model does not have access to the test set of FFHQ. We directly use the pre-trained DDPM by (Chung et al., 2022a), which is trained without the test set.

For PULSE (Menon et al., 2020) inversion of StypleGAN, we follow the original paper to run spherical gradient descent with learning rate $0.4$. We run gradient ascent for $500$ steps. For ILO (Daras et al., 2021) inversion of StyleGAN, we follow the original paper to run spherical gradient descent on $4$ different layers of StyleGAN with learning rate $0.4$. We run gradient ascent for $200, 200, 100, 100$ steps for each layer. For MCG (Chung et al., 2022b) and DPS (Chung et al., 2022a), we follow the original paper to run gradient descent for $1000$ steps, with scale parameter $\zeta$ increasing as bitrate increases. We explain in detail about how to select $\zeta$ in next section.

## B.2 ADDITIONAL ABLATION STUDY

**Why some inversion approaches fail** In Tab. 1, we show that some inversion approaches fail for image codec, even with our theoretical guarantee. This is because our theory relies on the assumption that the generative model perfectly learns the natural image distribution, while this is hard to achieve in practice. And we think this is where the gap lies. More specifically, GAN has better "precision" and worse "recall", while diffusion has a fair "precision" and fair "recall". The "precision" and "recall" is defined by Sajjadi et al. (2018). Or to say, the distribution learned by GAN largely lies in the natural image distribution better, but many area of natural image distribution is not covered. The distribution learned by diffusion does not lies in natural image distribution as good as GAN, but it covers more area of natural image distribution. This phenomena is also observed and discussed by Dhariwal & Nichol (2021) and Ho & Salimans (2022). The "precision" is more important for sampling, but "recall" is more important for inversion. That is why the GAN based approach fails. The MCG fails probably because it is not well suited for non-linear problems.

$\zeta$ **Selection** The $\zeta$ parameter in DPS (Chung et al., 2022a) is like the learning rate parameter in PULSE (Menon et al., 2020) and ILO (Daras et al., 2021). It controls the strength of idempotence constraint. When $\zeta$ is too small, the perceptual reconstruction will deviate from the original image, and the MSE will go beyond the MSE upperbound. When $\zeta$ is too large, artefacts will dominate the image, and again, the MSE will go beyond the MSE upperbound.

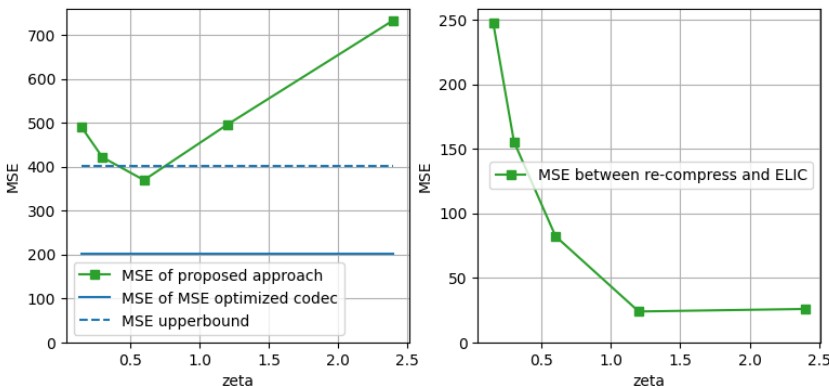

Figure 6: Ablation study on MSE $- \zeta$ with ImageNet dataset and ELIC. Left: MSE between source and reconstruction. Right: MSE between ELIC and re-compression.

As shown in Fig. 6, using a $\zeta$ too small or too large will cause the reconstruction MSE higher than the theoretical upperbound. While using a proper $\zeta$ achieves a MSE satisfies the MSE upperbound. On the other hand, the MSE of ELIC and re-compressed reconstruction indeed goes lower as $\zeta$ goes up. This is because this term is exactly what $\zeta$ is penalizing. Despite a large $\zeta$ strengthen the idempotence constraint, it can also push reconstruction off the learned natural image manifold. Another example is shown in Fig. 8. It is shown that using insufficiently large $\zeta$ can sometimes lead to weird reconstruction.

To further understand this phenomena, we visualize the perceptual reconstruction and the re-compression results of those perceptual reconstruction. As shown in Fig. 7, when $\zeta = 0.15, 0.3$, the re-compression image does not look like the ELIC reconstruction. And this indicates that the idempotence constraint is not strong enough. While when $\zeta = 0.6, 1.2, 2.4$, the re-compression image looks the same as ELIC reconstruction. However, when $\zeta = 1.2, 2.4$, the perceptual reconstruction is dominated by noise. And therefore its MSE still exceeds the MSE upperbound.

In practical implementation, we search $\zeta$ from $0.3$ and increase it when it is not large enough. For five bitrate of Hyper based model, we select $\zeta = \{0.3, 0.6, 1.2, 1.6, 1.6\}$ on FFHQ, $\{0.3, 0.6, 0.6, 1.2, 1.2\}$ on ImageNet, $\{0.6, 0.6, 0.6, 1.2, 1.2\}$ on COCO and $\{0.45, 0.9, 0.9, 1.2, 1.6\}$. For five bitrate of ELIC based model, we select $\zeta = \{0.3, 0.6, 1.2, 1.6, 1.6\}$ on FFHQ, $\{0.3, 0.6, 0.6, 1.2, 1.6\}$ on ImageNet, $\{0.6, 0.6, 0.6, 1.2, 1.2\}$ on COCO and $\{0.45, 0.6, 0.6, 1.2, 1.6\}$ on CLIC. Sometimes the $\zeta$ required for different images are different. Therefore, we also introduce an very simple adaptive $\zeta$ mechanics. More specifically, we

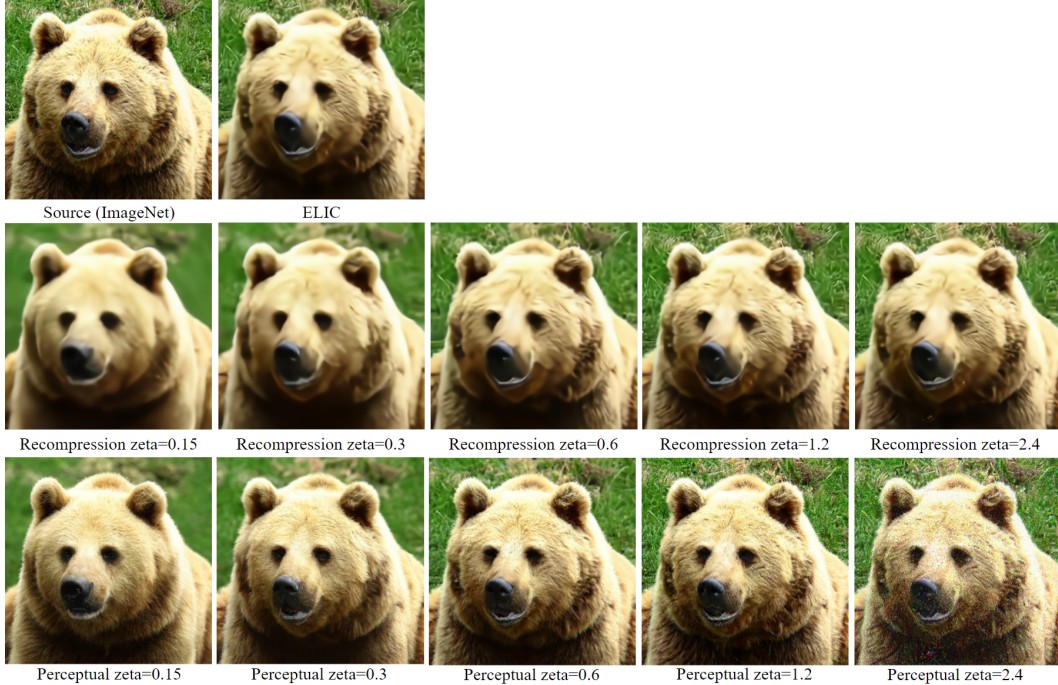

Figure 7: Ablation study on adjusting $\zeta$ with ImageNet dataset and ELIC.

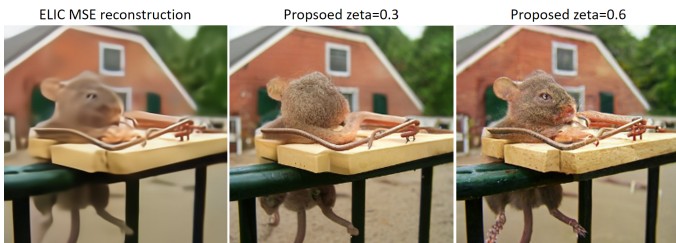

Figure 8: An example of using improper $\zeta$ with ImageNet dataset and ELIC.

evaluate the MSE between re-compressed perceptual reconstruction and MSE reconstruction. If it is larger than 32, we multiple $\zeta$ by 1.5 until $K$ time. We set $K = \{1, 1, 2, 4, 4\}$ for all methods and datasets. We note that it is possible to exhaustively search $\zeta$ for better perceptual quality. However, it will drastically slow down the testing.

## B.3 ADDITIONAL RESULTS

**Other Metrics** Besides the FID, PSNR and MSE, we also test our codec using other metrics such as Kernel Inception Distance (KID) (Binkowski et al., 2018) and LPIPS (Zhang et al., 2018). Similar to FID and PSNR, we report BD metrics:
The result of KID has same trend as FID, which means that our approach is sota. While many other approaches outperform our approach in LPIPS. This result is expected, as:

- KID is a divergence based metric. And our approach is optimized for divergence.

- LPIPS is a image-to-image distortion. By distortion-perception trade-off (Blau & Michaeli, 2018), perception optimized approaches can not achieve SOTA LPIPS.

- All other perceptual codec (HiFiC, ILLM, CDC, Po-ELIC) except for ours use LPIPS as loss function during training.

| Method | FFHQ | | ImageNet | | COCO | | CLIC | |
|---|---|---|---|---|---|---|---|---|
| | BD-logKID ↓ | BD-LPIPS ↓ | BD-logKID ↓ | BD-LPIPS ↓ | BD-logKID ↓ | BD-LPIPS ↓ | BD-logKID ↓ | BD-LPIPS ↓ |
| *MSE Baselines* | | | | | | | | |
| Hyper | 0.000 | 0.000 | 0.000 | 0.000 | 0.000 | 0.000 | 0.000 | 0.000 |
| ELIC | -0.232 | -0.040 | -0.348 | -0.058 | -0.236 | -0.062 | -0.406 | -0.059 |
| BPG | 0.1506 | -0.010 | 0.027 | -0.010 | 0.126 | -0.012 | 0.039 | -0.008 |
| VTM | -0.232 | -0.031 | -0.298 | -0.048 | -0.216 | -0.050 | -2.049 | -0.048 |
| *Conditional Generative Model-based* | | | | | | | | |
| HiFiC | -3.132 | -0.108 | -2.274 | -0.172 | -2.049 | -0.172 | -1.925 | -0.148 |
| HiFiC* | -4.261 | -0.110 | -2.780 | -0.173 | - | - | - | - |
| Po-ELIC | -3.504 | -0.104 | -2.877 | -0.167 | -2.671 | -0.168 | -2.609 | -0.145 |
| CDC | -2.072 | -0.060 | -1.968 | -0.099 | -1.978 | -0.101 | -2.122 | -0.084 |
| ILLM | -3.418 | -0.109 | -2.681 | -0.181 | -2.620 | -0.180 | -2.882 | -0.155 |
| ILLM* | -4.256 | -0.106 | -2.673 | -0.178 | - | - | - | - |
| *Unconditional Generative Model-based* | | | | | | | | |
| Proposed (Hyper) | -5.107 | -0.086 | -4.271 | -0.058 | -4.519 | -0.083 | -3.787 | -0.056 |
| Proposed (ELIC) | **-5.471** | -0.099 | **-5.694** | -0.106 | **-5.360** | -0.113 | **-4.046** | -0.079 |

Table 7: Results on FFHQ, ImageNet, COCO and CLIC. *: re-trained on corresponding dataset. **Bold**: lowest KID. Underline: second lowest KID.

| | FFHQ | ImageNet | COCO | CLIC |
|---|---|---|---|---|
| Hyper | 0.0000 | 0.0000 | 0.0000 | 0.0000 |
| ELIC | 0.0098 | 0.0148 | 0.0146 | 0.0176 |
| HiFiC | 0.0030 | 0.0064 | 0.0071 | 0.0058 |
| ILLM | 0.0001 | 0.0041 | 0.0058 | 0.0042 |
| Proposed (Hyper) | -0.0065 | -0.0399 | -0.0262 | -0.0236 |
| Proposed (ELIC) | -0.0029 | -0.0203 | -0.0135 | -0.0132 |

Table 8: MS-SSIM on FFHQ, ImageNet, COCO and CLIC.

- ILLM (Muckley et al., 2023) also achieves sota FID, KID and visual quality at that time, but its LPIPS is outperformed by an autoencoder trained with LPIPS.

In terms of MS-SSIM, ELIC > HiFiC > ILLM > Hyper > Proposed (ELIC) > Proposed (Hyper). We are reluctant to use MS-SSIM as perceptual metric, as this result is obviously not aligned with visual quality, and should not be used when we considering divergence based perceptual quality as (Blau & Michaeli, 2018), because:

- MS-SSIM is an image to image distortion. By Blau & Michaeli (2018), it is in odd with divergence based metrics such as FID, KID. Theoretically, there does not exist a codec that achieve optimal FID and MS-SSIM at the same time.

- In terms of MS-SSIM, ELIC, a mse optimzied codec, is the sota. This indicates that MS-SSIM correlates poorly with human perceptual quality.

- Similar result is also reported by Muckley et al. (2023) Fig. 3, where the MS-SSIM of ILLM is not even as good as Hyper, which is a mse optimized codec. This indicates that MS-SSIM correlates poorly with human perceptual quality.

- In a perceptual codec competition CVPR CLIC 2021, the human perceptual quality is tested as final result and MS-SSIM, FID are evaluated. The 1st place of human perceptual test result among 23 teams "MIATL_NG", also has lowest FID, which indicates FID correlates well with human perceptual quality. On the other hand, its MS-SSIM ranks 21st place among 23 teams, which indicates MS-SSIM correlates poorly with human perceptual quality.

**Rate Distortion Curve** We do not have enough room in the main text to show rate-distortion curve. We present them in Fig. 11 of appendix instead.

**Perception Distortion Trade-off** A couple of perception image codec have achieved perception-distortion trade-off with the same bitstream (Iwai et al., 2020; Agustsson et al., 2022; Goose et al.,

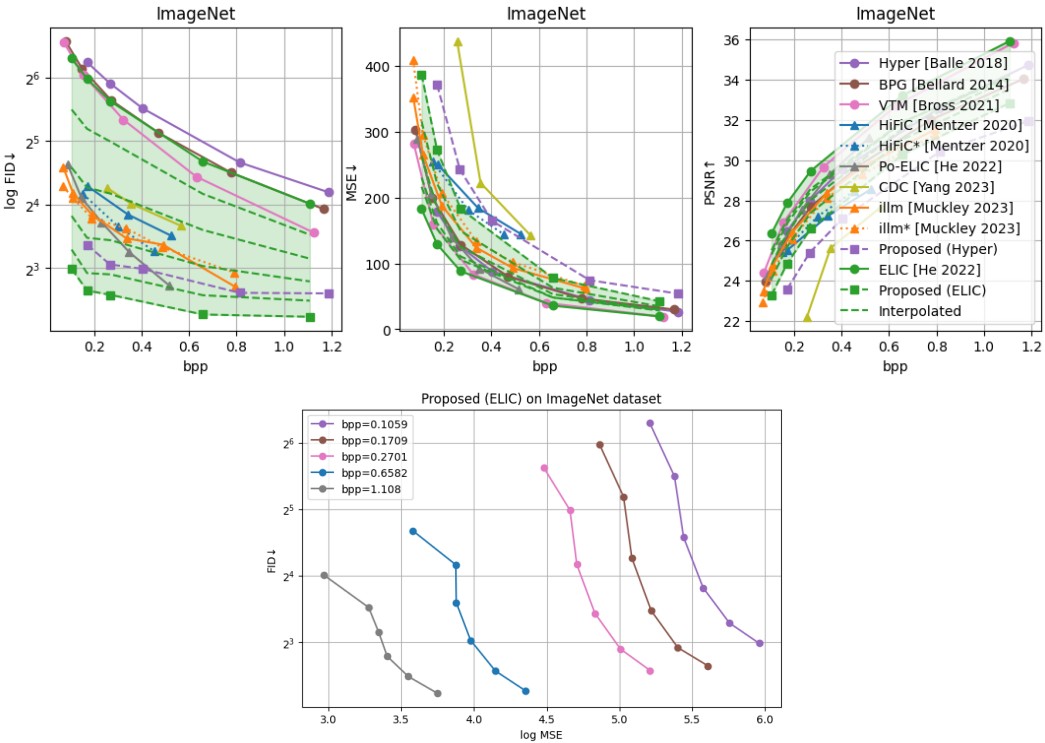

Figure 9: Perception distortion trade-off by convex interpolation.

2023). As our proposed approach shares the same bitstream as the MSE codec, we are already capable of decoding the optimal perceptual quality and optimal MSE at the same time. To achieve the intermediate points of perception-distortion trade-off, we can utilize the convex interpolation method proposed by Yan et al. (2022). More specifically, Yan et al. (2022) prove that when divergence is measured in Wasserstein-2 (W2) distance, convex combination of perceptual and MSE reconstruction is optimal in distortion-perception trade-off sense. In our case, even if the divergence is not W2, convex combination is able to achieve a reasonable result. More specifically, we denote the MSE reconstruction as $\hat{X}_\Delta$, and perceptual reconstruction as $\hat{X}_p$. The convex combination is

$$\hat{X}_\alpha = \alpha \hat{X}_p + (1 - \alpha)\hat{X}_\Delta, \alpha \in [0.0, 1.0]. \tag{25}$$

We evaluate this approach on ImageNet dataset with ELIC as base codec, as this setting covers the widest range of perception and distortion. In Fig. 9, we can see that our interpolated codec successfully cover a wide range of perception-distortion trade-off, More specifically, our optimal perception codec has FID lower than ILLM (Muckley et al., 2023) but a MSE higher than ILLM. However, the interpolation with $\alpha = 0.8$ still has a FID lower than ILLM. But its MSE is already on par with ILLM. And the interpolation with $\alpha = 0.6$ has a FID comparable to ILLM, but its MSE is obviously lower. This indicates that the interpolated version of our codec remains competitive in perception-distortion trade-off.

**Visual Comparison of Different Bitrate** We visualize our reconstruction from low to high bitrate, along with ELIC as our bitrate is the same. It is interesting to find that both our approach and ELIC converge to the source image as bitrate grows high, but from different directions. For us, the visual quality does not differ much as bitrate goes high, while the reconstruction becomes more aligned with the source. For ELIC, the reconstruction grows from blurry to sharp as bitrate goes high.

**Visual Comparison with Other Methods** We do not have enough room in the main text to compare all approaches visually. In main text Fig. 4, we only compare our approach to HiFiC (Mentzer et al., 2020) and ILLM (Muckley et al., 2023). Thus, we put the visual results of other methods in Fig. 12-15 here in Appendix.

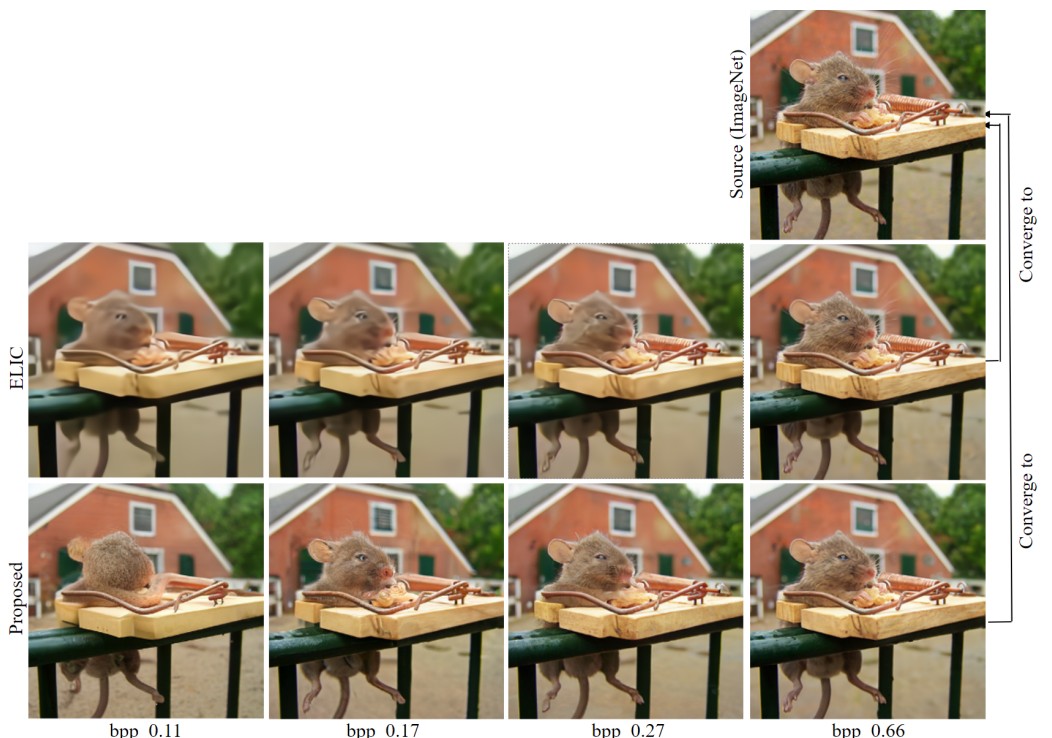

Figure 10: The visual result of ELIC and our proposed approach as bitrate goes high gradually.

**Reconstruction Diversity** In main text Fig. 5, we only present three alternative reconstructions that is used to compute standard deviation. In Fig. 16, we present additional 12 reconstructions to show that our reconstruction has good diversity.

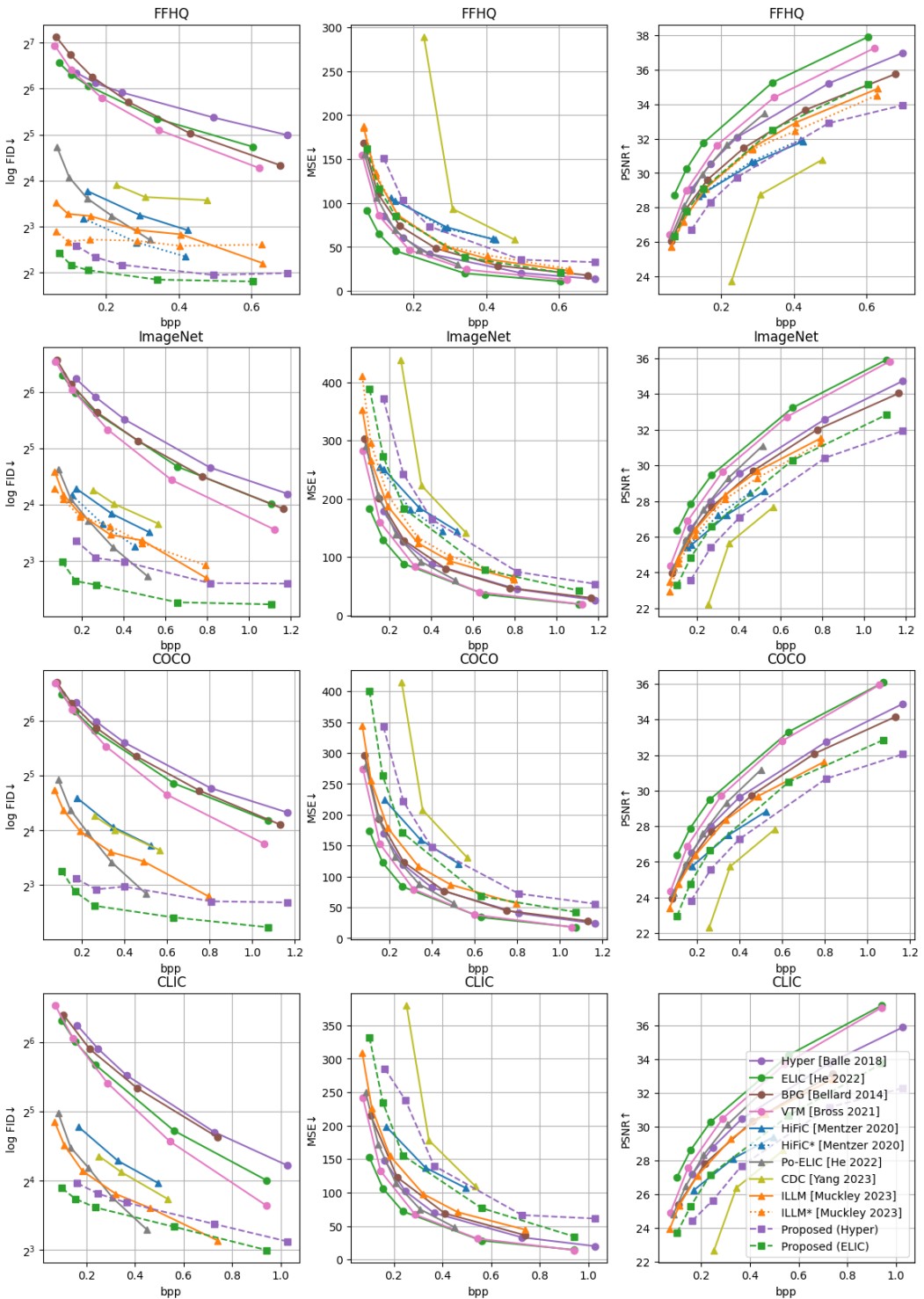

Figure 11: The rate-distortion curve of different methods.

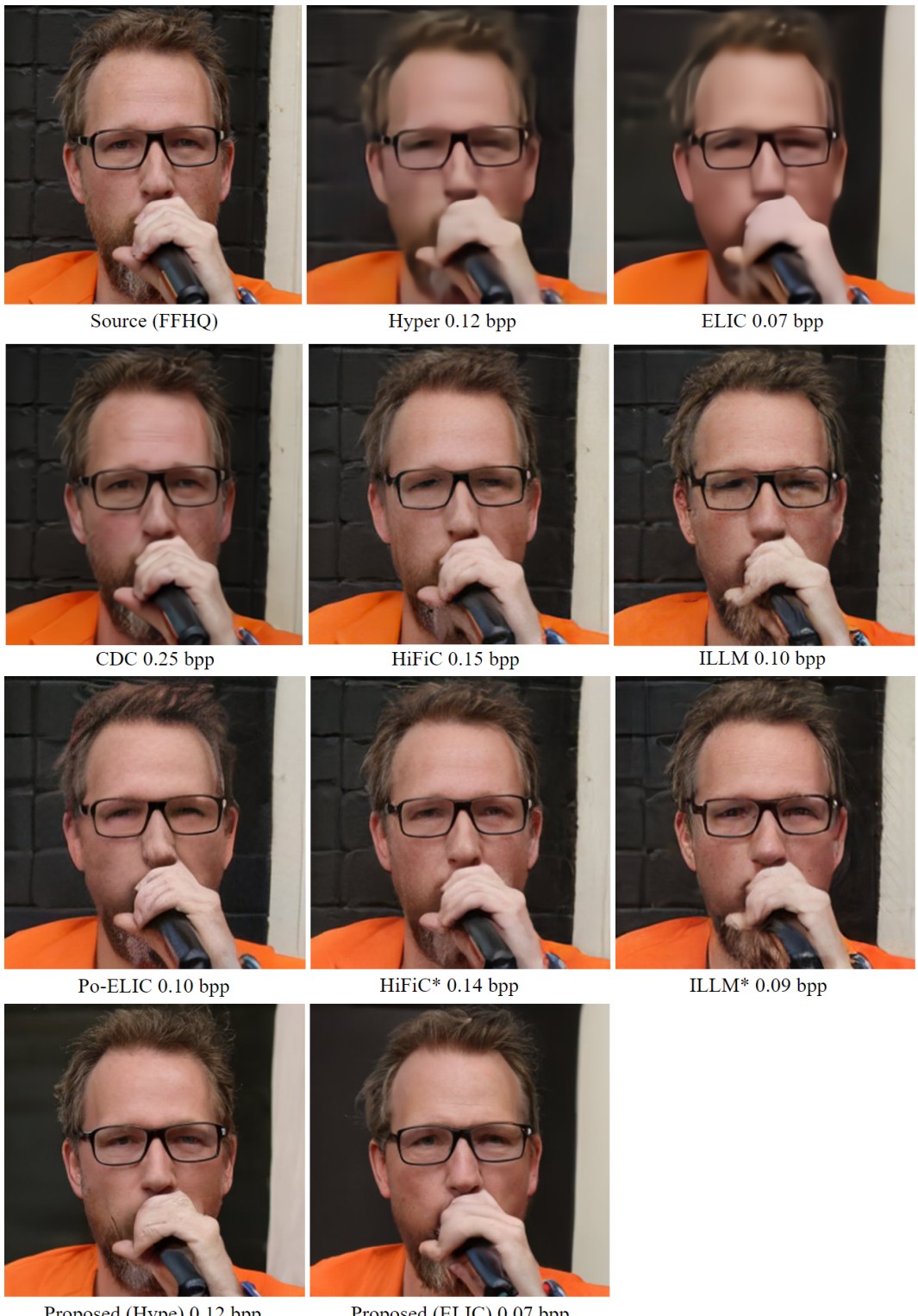

Figure 12: A visual comparison of our proposed approach with other approaches.

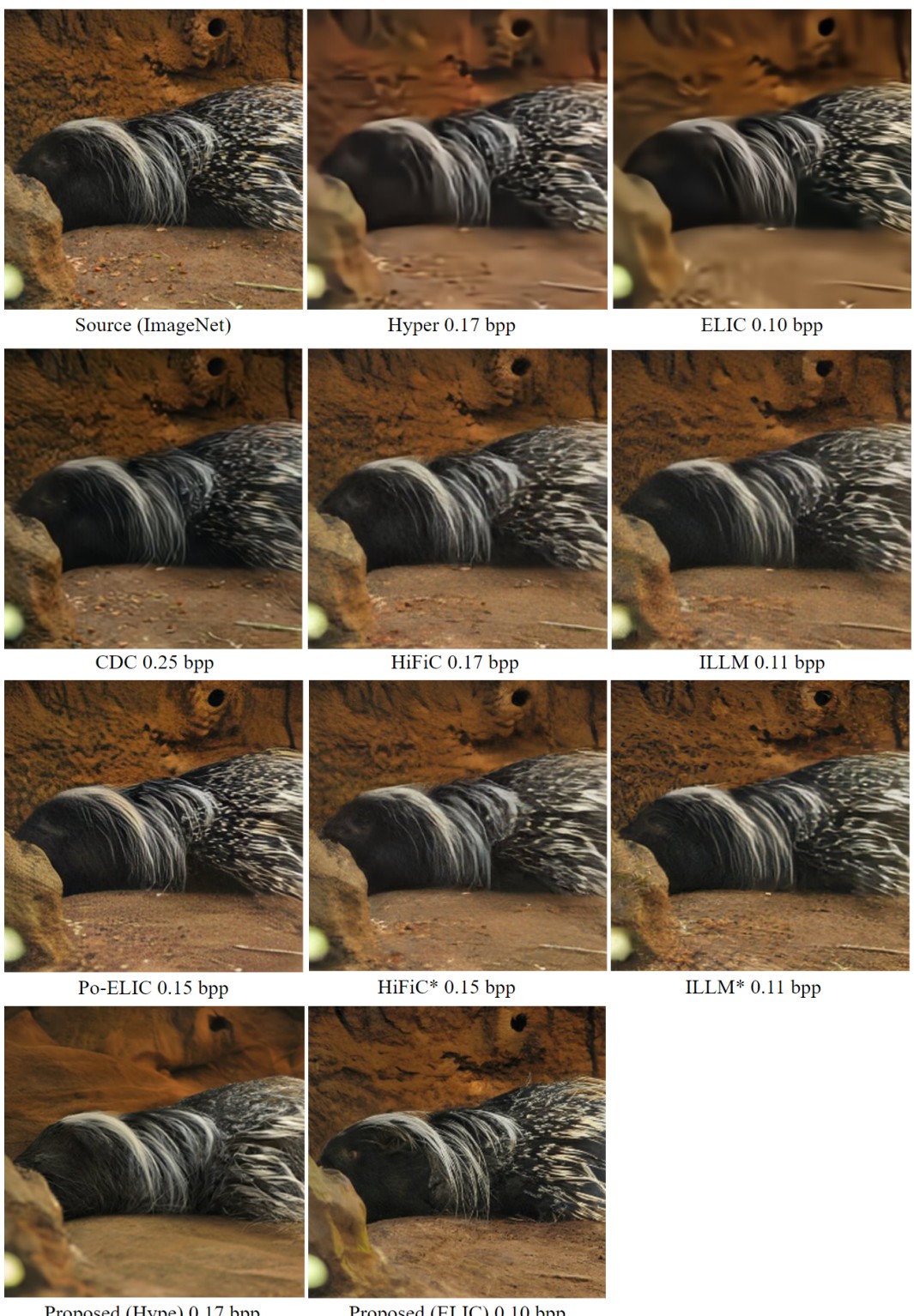

Figure 13: A visual comparison of our proposed approach with other approaches.

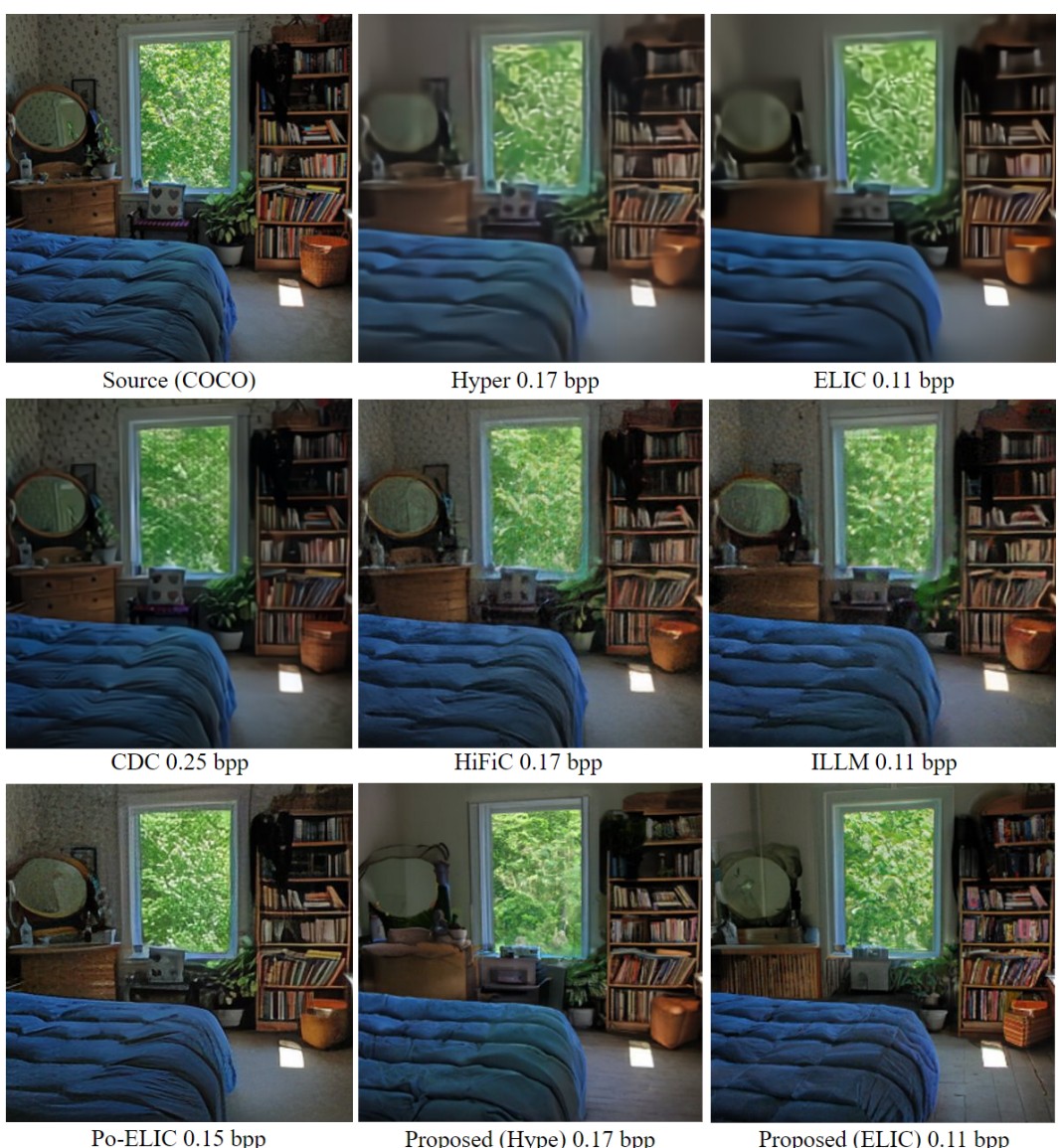

Figure 14: A visual comparison of our proposed approach with other approaches.

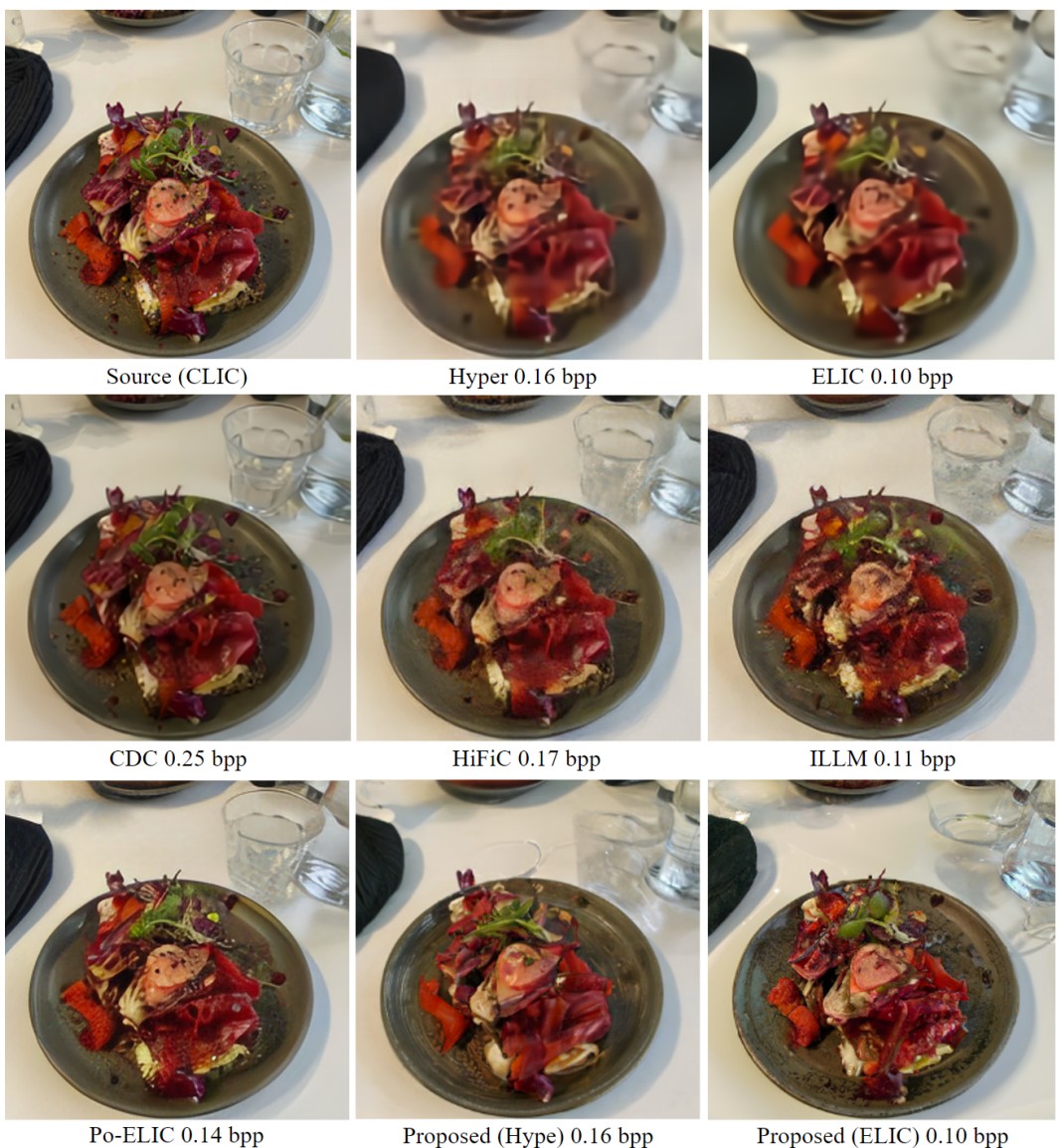

Figure 15: A visual comparison of our proposed approach with other approaches.

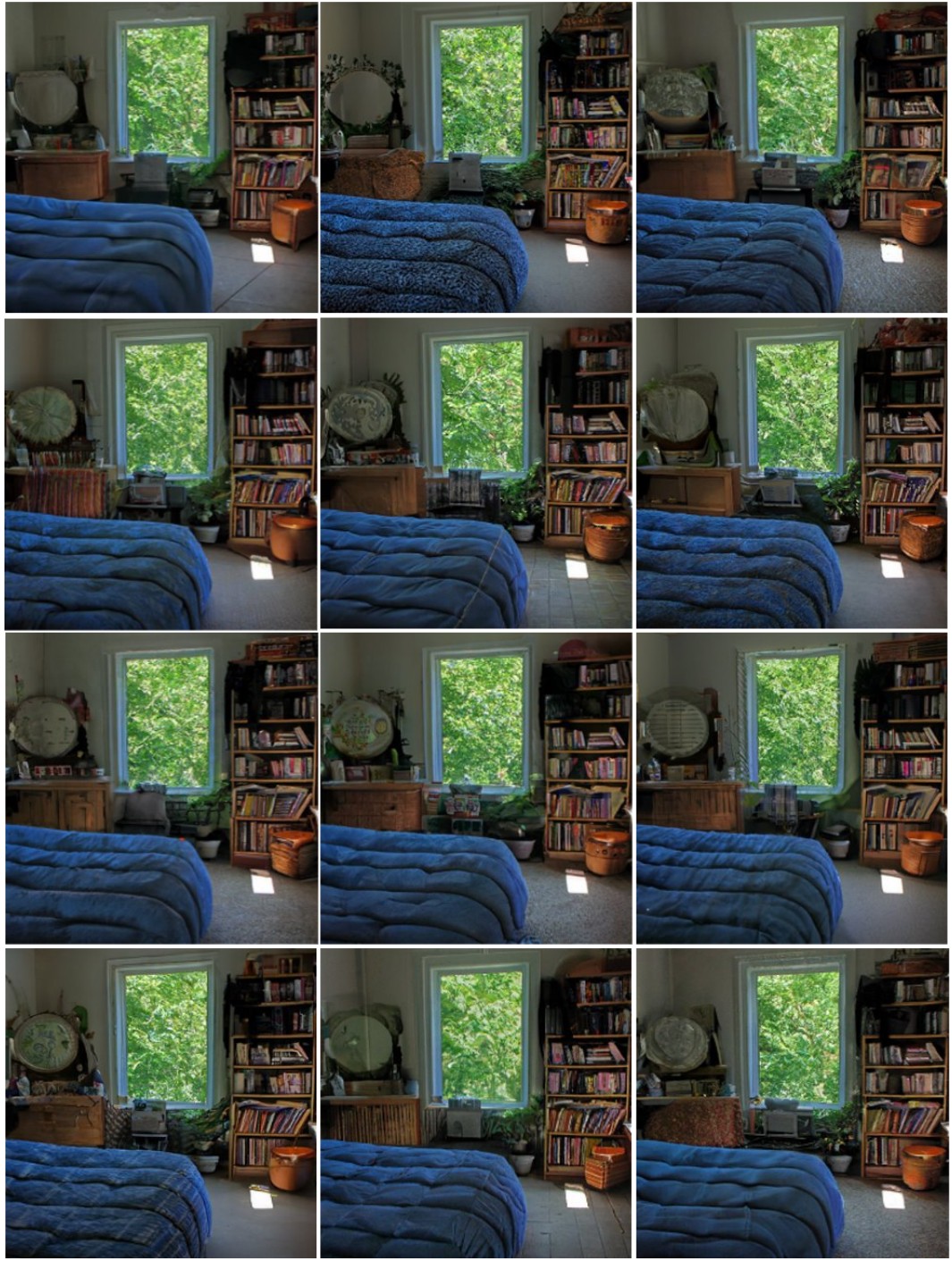

Figure 16: Reconstruction diversity of proposed approach.

