# OpenReview forum: "Idempotence and Perceptual Image Compression"
_ICLR.cc/2024/Conference — ICLR 2024 spotlight_

### Official Review · Reviewer_zF7s · 2023-10-28

**Soundness:** 3 good
**Presentation:** 2 fair
**Contribution:** 3 good
**Rating:** 6
**Confidence:** 3

**Summary:**

This text introduces a new concept of idempotence within image codec stability, revealing its unexpected relationship with perceptual image compression. By leveraging this understanding, the proposed method utilizes idempotence constraints to invert unconditional generative models, presenting an equivalent and improved paradigm for perceptual image codecs.

**Strengths:**

This paper presents a new paradigm of perceptual image codec, which could bring new insight to the community. The approach doesn't necessitate new model training but rather utilizes pre-trained mean-square-error codecs and unconditional generative models.

**Weaknesses:**

However, I also have some concerns as follows:

1) The relationship between idempotence and image compression is hard to understand in the current version.

2) Authors should provide more evidence to support the points that Idempotence brings perceptual quality in Section 3.

3) I can't see the superiority of the proposed method in the visual comparison with HiFiC and ILLM, especially in Fig 1.

4) In BD-FID, the proposed method is better. But in BD-PSNR, others may be better. Could authors provide more evaluation metrics, such as MSSSIM and VMAF.

**Questions:**

See weaknesses.

---

> ### Author Response · Authors · 2023-11-13
> **Rebuttal by Authors Part I**
>
> Thanks for your detailed review. And we are glad to provide our answer to your questions:
>
> * __W1__. Although we have provided proof sketch in main text and complete proof in Appendix of __Theorem 1__ and __Theorem 2__, we do agree that they are quite abstract and not intuitive. However, theoretical results take $50$ percent of this paper. And it is a must to understand them correctly to understand this paper. To provide a more intuitive illustration, we will include an example with discrete finite alphabet (our theoretical results are appliable to any countable alphabet or Borel-measurable source $X$ and any deterministic measureable function $f(.)$):
>     * __Example 1__ (1-dimension, discrete finite alphabet, optimal 1-bit codec): Consider discrete 1d random variable $X$, with alphabet $\mathcal{X} = \{0,1,2,3,4,5\}$, and $Y$ with alphabet $\mathcal{Y}=\{0,1\}$. We assume $P(X=i) = \frac{1}{6}$, i.e., $X$ follows uniform distribution. We consider a deterministic transform $f(X): \mathcal{X}\rightarrow \mathcal{Y} = \textrm{round}(X/3)$. Or to say, $f(.)$ maps $\{0,1,2\}$ to $\{0\}$, and $\{3,4,5\}$ to $\{1\}$. This is infact the MSE-optimal 1-bit codec for source $X$ (See Chapter 10 of [Elements of Information Theory]). And the joint distribution $P(X,Y)$, posterior $P(X|Y)$ is just a tabular
>     *   | x | y | P(X=x,Y=y) | P(X=x\|Y=y) | f(X=x) |
>         |---|---|------------|-------------|--------|
>         | 0 | 0 | 1/6        | 1           | 0      |
>         | 1 | 0 | 1/6        | 1           | 0      |
>         | 2 | 0 | 1/6        | 1           | 0      |
>         | 3 | 0 | 0          | 0           | 1      |
>         | 4 | 0 | 0          | 0           | 1      |
>         | 5 | 0 | 0          | 0           | 1      |
>         | 0 | 1 | 0          | 0           | 0      |
>         | 1 | 1 | 0          | 0           | 0      |
>         | 2 | 1 | 0          | 0           | 0      |
>         | 3 | 1 | 1/6        | 1           | 1      |
>         | 4 | 1 | 1/6        | 1           | 1      |
>         | 5 | 1 | 1/6        | 1           | 1      |
>     * We first examine __Theorem 1__, which says that any sample from $X\sim P(X|Y=y)$ satisfies $f(X) = y$. Observing this table, this is indeed true. As for $f(X)\neq y$, the posterior $P(X=x|Y=y)$ is $0$.
>     * We then examine __Theorem 2__, which says that sampling from $X\sim P(X)$ with $f(X)=y$ constraint is the same as sampling from $P(X|Y=y)$. We first identify the set such that $f(X)=y$. When $y=0$, this set is $\{0,1,2\}$. And when $y=1$, this set is $\{3,4,5\}$. And sampling from $p(X)$ with constrain $f(X)=y$ is equvalient to sampling from one of the two subset with probability $\propto p(X)$. And this probability is exactly $P(X|Y=y)$.
> * __W2__. The "Idempotence brings perceptual quality" in Sec. 3 is __Theorem 2__. It is already theoretically supported by proof in Appendix. A. Furthermore, it is already empirically supported by the Sec. 5. In Tab. 2 and Fig. 5 of Sec. 5, we can clearly see that indeed optimizing idempotence loss brings perceptual image compression.
> * __W3__. It could be the problem of PDF compression. We have provided a version with larger resolution in https://ibb.co/NWww3RX. Probably zoom in can help (Reviewer VEis already finds qualitative results compelling with the PDF). We note that only our approach can reconstruct the beard of dog, while HiFiC and ILLM fail to do so. Further, our reconstruction has much more realistic texture of dog's hair.

---

> ### Author Response · Authors · 2023-11-13
> **Rebuttal by Authors Part II**
>
> * __W4__. For BD-PSNR, we emphasis that:
>     * We can control our decoder to achieve an intermediate point between FID and MSE. In Appendix B.3 and Fig. 8, we can see that by trading FID for MSE, our approach achieves better FID with comparable MSE with ILLM. And it can also achieves comparable FID with better MSE compared with ILLM. We also note that it is impossible to achieve sota FID and sota PSNR at the same time by distortion-perception trade-off [Blau 2018]. Therefore, we can not achieve a PSNR as good as [Balle 2018] and [He 2022].
>     * Our MSE is within 2x MSE of base codec, and our PSNR is within 3 dB of base codec, which is aligned to the theoretical result.
> * For other metrics, we have evaluated our approach additionally on KID and LPIPS:
>
>     | Method              | FFHQ      |          | ImageNet  |          | COCO      |          | CLIC      |          |
>     |---------------------|-----------|----------|-----------|----------|-----------|----------|-----------|----------|
>     |                     | BD-logKID↓| BD-LPIPS↓| BD-logKID↓| BD-LPIPS↓| BD-logKID↓| BD-LPIPS↓| BD-logKID↓| BD-LPIPS↓|
>     | _MSE Baselines_     |           |          |           |          |           |          |           |          |
>     | Hyper               | 0.0000    | 0.0000   | 0.0000    | 0.0000   | 0.0000    | 0.0000   | 0.0000    | 0.0000   |
>     | ELIC                | -0.2320   | -0.0402  | -0.348    | -0.0580  | -0.236    | -0.0623  | -0.4067   | -0.0595  |
>     | BPG                 | 0.1506    | -0.0102  | 0.0274    | -0.0102  | 0.1267    | -0.0116  | 0.03985   | -0.00795 |
>     | VTM                 | -0.2320   | -0.0311  | -0.2975   | -0.0476  | -0.2157   | -0.0504  | -2.049    | -0.04812 |
>     | _Conditional-based_ |           |          |           |          |           |          |           |          |
>     | HiFiC               | -3.132    | -0.1087  | -2.274    | -0.1724  | -2.049    | -0.1723  | -1.925    | -0.1483  |
>     | HiFiC*              | -4.261    | -0.1102  | -2.780    | -0.1734  | N/A       | N/A      | N/A       | N/A      |
>     | Po-ELIC             | -3.504    | -0.1047  | -2.877    | -0.1674  | -2.671    | -0.1687  | -2.609    | -0.145   |
>     | CDC                 | -2.072    | -0.0600  | -1.968    | -0.0985  | -1.978    | -0.1011  | -2.122    | -0.08437 |
>     | ILLM                | -3.418    | -0.1092  | -2.681    | -0.1809  | -2.62     | -0.1805  | -2.882    | -0.1547  |
>     | ILLM*               | -4.256    | -0.1062  | -2.673    | -0.1784  | N/A       | N/A      | N/A       | N/A      |
>     |_Unconditional-based_|           |          |           |          |           |          |           |          |
>     | Proposed (Hyper)    | -5.107    | -0.0859  | -4.271    | -0.0576  | -4.519    | -0.0826  | -3.787    | -0.0557  |
>     | Proposed (ELIC)     | -5.471    | -0.0987  | -5.694    | -0.1062  | -5.36     | -0.113   | -4.046    | -0.0796  |
> * The result of KID has same trend as FID, which means that our approach is sota. While many other approaches outperform our approach in LPIPS. This result is expected, as:
>     * KID is a divergence based metric. And our approach is optimized for divergence.
>     * LPIPS is a image-to-image distortion. By distortion-perception trade-off [Blau 2018], perception optimized approaches can not achieve SOTA LPIPS.
>     * All other perceptual codec (HiFiC, ILLM, CDC, Po-ELIC) except for ours use LPIPS as loss function during training.
>     * ILLM [Muckley 2023] also achieves sota FID, KID and visual quality at that time, but its LPIPS is outperformed by a autoencoder trained with LPIPS.

---

> ### Comment · Reviewer_zF7s · 2023-11-21
>
> Thank the authors for the reply. I partly insist on my point that I can't see the superiority of the proposed method in the visual comparison with HiFiC and ILLM without the comparison of MS-SSIM. If the authors want to change my point, please show those real results to me. Even if the results are not better than HiFIC and ILLM, I will increase my score.

---

> ### Author Response · Authors · 2023-11-21
>
> Sure, below we present the result of MS-SSIM:
>
> |                  | FFHQ    | ImageNet | COCO    | CLIC    |
> |------------------|---------|----------|---------|---------|
> | Hyper            | 0.0000  | 0.0000   | 0.0000  | 0.0000  |
> | ELIC             | 0.0098  | 0.0148   | 0.0146  | 0.0176  |
> | HiFiC            | 0.0030  | 0.0064   | 0.0071  | 0.0058  |
> | ILLM             | 0.0001  | 0.0041   | 0.0058  | 0.0042  |
> | Proposed (Hyper) | -0.0065 | -0.0399  | -0.0262 | -0.0236 |
> | Proposed (ELIC)  | -0.0029 | -0.0203  | -0.0135 | -0.0132 |
>
> In terms of MS-SSIM, ELIC > HiFiC > ILLM > Hyper > Proposed (ELIC) > Proposed (Hyper). We are reluctant to use MS-SSIM as perceptual metric, as this result is obviously not aligned with visual quality, and should not be used when we considering divergence based perceptual quality as [Blau 2018], because:
> * MS-SSIM is an image to image distortion. By [Blau 2018], it is in odd with divergence based metrics such as FID, KID. Theoretically, there does not exist a codec that achieve optimal FID and MS-SSIM at the same time.
> * In terms of MS-SSIM, ELIC, a mse optimzied codec, is the sota. This indicates that MS-SSIM correlates poorly with human perceptual quality.
> * Similar result is also reported in [Muckley 2023] Fig. 3, where the MS-SSIM of ILLM is not even as good as Hyper, which is a mse optimized codec. This indicates that MS-SSIM correlates poorly with human perceptual quality.
> * In a perceptual codec competition CVPR CLIC 2021 (https://clic.compression.cc/2021/leaderboard/image-075/test/index.html), the human perceptual quality is tested as final result and MS-SSIM, FID are evaluated. The 1st place of human perceptual test result among 23 teams "MIATL_NG", also has lowest FID, which indicates FID correlates well with human perceptual quality. On the other hand, its MS-SSIM ranks 21st place among 23 teams, which indicates MS-SSIM correlates poorly with human perceptual quality.
>
> We will include MS-SSIM result in our final version, but we emphasis that this result should be treated with care.

---

> > ### Comment · Reviewer_zF7s · 2023-11-21
> >
> > Thanks for the prompt response.  I find the results satisfactory. I've managed to improve my score, and while I acknowledge that the proposed method may not outperform in the MS-SSIM metric, I trust that the authors will incorporate all the updated MS-SSIM results in the revised paper. In my perspective, I consider MS-SSIM a metric that effectively reflects structural quality.

---

### Official Review · Reviewer_VEis · 2023-10-30

**Soundness:** 4 excellent
**Presentation:** 3 good
**Contribution:** 4 excellent
**Rating:** 8
**Confidence:** 4

**Summary:**

The paper proposes a theoretical justification for consideration of idempotence in the task of generative image compression. This includes a couple of theoretical results: the first is that a perceptual codec is idempotent, and the second is that an MSE-optimal codec paired with an unconditional generative model is optimal in terms of rate-distortion-perception theory. The paper includes a suite of empirical studies to justify the theory, showing that the inclusion of an idempotence constraint in the sampling process of a generative model (on top of a pretrianed MSE codec) gives better rate-distortion-perception performance than the previous methods considered when using a DDPM+DPS diffusion model sampler.

**Strengths:**

This paper makes contributions on a number of areas to the field of generative image compression.

1. It presents a theoretical justification for why generative codecs should be idempotent.
2. It presents a theoretical justification for why an MSE-optimal codec paired with an unconditional model. This is an approach considered by other works in the field such as (Hoogeboom, 2023) and (Ghouse, 2023).
3. Empirical results further justify the insights of theory, as the presented method is optimal among all generative codecs considered.
4. Empirical results include ablations over the base MSE autoencoder, which allows fair comparisons to both the ILLM and HiFiC methods of previous work (which use older autoencoders).
5. Qualitative results of the images are compelling.

**Weaknesses:**

1. The primary weakness is a lack of comparison to other diffusion-based methods. The only diffusion method considered is CDC, which may be relatively underpowered vs. DIRAC (Ghouse, 2023). CDC is almost certainly underpowered vs. the work of (Hoogeboom, 2023). Also, there are images released from the work of (Agustsson, 2022), which are not included in the paper.
2. The x-domain constraint seems identical to that of (Hoogeboom, 2023).
3. The paper uses a third-party implementation of HiFiC with a different training set that may give divergent results from the original paper. The authors of the HiFiC paper have made their models available at https://github.com/tensorflow/compression.
4. Only two metrics are considered: FID and PSNR. Most other compression works include more metrics, such as LPIPS, to gauge model performance across a variety of axes.

The proposed method uses much more compute than its competitors, and it has limited image resolutions. However, I am not counting this is as a negative for my review, because this is clearly stated already in the manuscript.

**Questions:**

1. The failures of many of the generative models to give good results suggests a divergence between theory in practice, i.e., although the theory suggests any unconditional generator will do, the generators that we have available to us and their various samplers exhibit widely different properties. Did the authors consider discussing this in the Discussion?
2. Could you clarify why you used the OASIS FID implementation as opposed to the FID/256 method that is more standard in the compression community?
3. The rate-distortion-perception tradeoff is portrayed in a confusing way in Figure 8 of the Appendix. Is the reason that you could not match bpps between methods as done in Figure 1 of (Muckley, 2023)?
4. Would you consider to use language other than "perception" to describe distributional divergence? "Perception" is an overloaded term in the compression community and does not intuitively describe the phenomena of the paper. Although it is true that "perception" is the term of Blau/Michaeli to describe divergence phenomena, others in the community have adopted more specific teams such as "realism" or "statistical fidelity" that lead to less confusion with human perception.

---

> ### Author Response · Authors · 2023-11-13
> **Rebuttal by Authors Part I**
>
> Thanks for your detailed review. And we are glad to provide our answer to your questions:
>
> * __W1__. The reason why we only compare to CDC among diffusion codec is that it is the only diffusion codec that provides test code. Furthermore, [Goose 2023] [Hoogeboom 2023] are not yet published. Not comparing to them should not be considered as a weakness. Our testing dataset has different size and resolution than [Agustsson 2022], and that is why we have not compared to the image released. However, we do note that [Muckley 2023] outperform the datapoint released by [Agustsson 2022] in terms of FID, and we outperform [Muckley 2023] in terms of FID.
> * __W2__. We are not sure if we get this correctly, so please correct us if our understanding is wrong. It appears to us that [Hoogeboom 2023] is a conditional generative codec, while its condition is MSE reconstruction $g_0(f_0(x))$. The diffusion part takes $g_0(f_0(x))$ as condition, and rectified flow allows very fast sampling within a few steps. We search for the word "unconditional" in [Hoogeboom 2023] but we fail to find relevant information. Thus it appears to me that [Hoogeboom 2023] is a conditional codec using $g_0(f_0(x))$ as condition, while x-domain constraint is $||g_0(f_0(\hat{x}))-g_0(f_0(x))||$. They both contains $g_0(f_0(x))$, but we do not think they are identical.
> * __W3__. Sure, we re-evaluate the original tensorflow HiFiC, and the results are as follows:
>
>     |                    | FFHQ   |         | ImageNet |         | COCO   |         | CLIC   |         |
>     |--------------------|--------|---------|----------|---------|--------|---------|--------|---------|
>     |                    | BD-FID↓| BD-PSNR↓| BD-FID↓  | BD-PSNR↓| BD-FID↓| BD-PSNR↓| BD-FID↓| BD-PSNR↓|
>     | HiFiC (Pytorch)    | -48.35 | -2.036  | -44.52   | -1.418  | -44.88 | -1.276  | -36.16 | -1.621  |
>     | HiFiC (Tensorflow) | -46.44 | -1.195  | -40.25   | -0.8287 | -46.45 | -0.9176 | -35.90 | -0.9202 |
>
> * In terms of BD-FID, the torch version outperforms tf version in FFHQ, ImageNet and CLIC dataset, while it is outperformed by tf version in COCO. The overall conclusion is not impacted.

---

> ### Author Response · Authors · 2023-11-13
> **Rebuttal by Authors Part II**
>
> * __W4__. Sure, we have evaluated our approach additionally on KID and LPIPS:
>
>     | Method              | FFHQ      |          | ImageNet  |          | COCO      |          | CLIC      |          |
>     |---------------------|-----------|----------|-----------|----------|-----------|----------|-----------|----------|
>     |                     | BD-logKID↓| BD-LPIPS↓| BD-logKID↓| BD-LPIPS↓| BD-logKID↓| BD-LPIPS↓| BD-logKID↓| BD-LPIPS↓|
>     | _MSE Baselines_     |           |          |           |          |           |          |           |          |
>     | Hyper               | 0.0000    | 0.0000   | 0.0000    | 0.0000   | 0.0000    | 0.0000   | 0.0000    | 0.0000   |
>     | ELIC                | -0.2320   | -0.0402  | -0.348    | -0.0580  | -0.236    | -0.0623  | -0.4067   | -0.0595  |
>     | BPG                 | 0.1506    | -0.0102  | 0.0274    | -0.0102  | 0.1267    | -0.0116  | 0.03985   | -0.00795 |
>     | VTM                 | -0.2320   | -0.0311  | -0.2975   | -0.0476  | -0.2157   | -0.0504  | -2.049    | -0.04812 |
>     | _Conditional-based_ |           |          |           |          |           |          |           |          |
>     | HiFiC               | -3.132    | -0.1087  | -2.274    | -0.1724  | -2.049    | -0.1723  | -1.925    | -0.1483  |
>     | HiFiC*              | -4.261    | -0.1102  | -2.780    | -0.1734  | N/A       | N/A      | N/A       | N/A      |
>     | Po-ELIC             | -3.504    | -0.1047  | -2.877    | -0.1674  | -2.671    | -0.1687  | -2.609    | -0.145   |
>     | CDC                 | -2.072    | -0.0600  | -1.968    | -0.0985  | -1.978    | -0.1011  | -2.122    | -0.08437 |
>     | ILLM                | -3.418    | -0.1092  | -2.681    | -0.1809  | -2.62     | -0.1805  | -2.882    | -0.1547  |
>     | ILLM*               | -4.256    | -0.1062  | -2.673    | -0.1784  | N/A       | N/A      | N/A       | N/A      |
>     |_Unconditional-based_|           |          |           |          |           |          |           |          |
>     | Proposed (Hyper)    | -5.107    | -0.0859  | -4.271    | -0.0576  | -4.519    | -0.0826  | -3.787    | -0.0557  |
>     | Proposed (ELIC)     | -5.471    | -0.0987  | -5.694    | -0.1062  | -5.36     | -0.113   | -4.046    | -0.0796  |
> * The result of KID has same trend as FID, which means that our approach is sota. While many other approaches outperform our approach in LPIPS. This result is expected, as:
>     * KID is a divergence based metric. And our approach is optimized for divergence.
>     * LPIPS is a image-to-image distortion. By distortion-perception trade-off [Blau 2018], perception optimized approaches can not achieve SOTA LPIPS.
>     * All other perceptual codec (HiFiC, ILLM, CDC, Po-ELIC) except for ours use LPIPS as loss function during training.
>     * ILLM [Muckley 2023] also achieves sota FID, KID and visual quality at that time, but its LPIPS is outperformed by a autoencoder trained with LPIPS.

---

> ### Author Response · Authors · 2023-11-13
> **Rebuttal by Authors Part III**
>
> * __Q1__. Thanks for the advice, we will add this discussion. Our theory relies on the assumption that the generative model perfectly learns the natural image distribution, while this is hard to achieve in practice. And we think this is where the gap lies. More specifically, GAN has better "precision" and worse "recall", while diffusion has a fair "precision" and fair "recall". The "precision" and "recall" is defined in [Improved precision and recall metric for assessing generative models]. Or to say, the distribution learned by GAN largely lies in the natural image distribution better, but many area of natural image distribution is not covered. The distribution learned by diffusion does not lies in natural image distribution as good as GAN, but it covers more area of natural image distribution. This phenomena is also observed and discussed by [Diffusion Models Beat GANs on Image Synthesis] and [Classifier-Free Diffusion Guidance]. The "precision" is more important for sampling, but "recall" is more important for inversion. That is why the GAN based approach fails. The MCG fails probably because it is not well suited for non-linear problems.
> * __Q2__. [Agustsson 2022] [Muckley 2023] evaluate FID with pathsize 256 on 256x256 COCO dataset with 30000 images and around 2000x2000 CLIC dataset with 400 images. The former produces 30000 patches and the later produces around 15000 pathes. As our approach is more expensive to evaluate, we can not afford testing on that large dataset. Therefore, we test our approach on 256x256 COCO dataset with 1000 images and 256x256 CLIC dataset with 400 images. In that case, using FID/256 will produce only 1000 and 400 patches. As FID is biased [Effectively Unbiased FID and Inception Score and where to find them], 1000 datapoints is too small for FID evaluation. Therefore, we evaluate FID with 64x64 patches, which produce 16000 and 6400 datapoints, which is reasonablely large. And this patch number is close to [Agustsson 2022] [Muckley 2023]. Note FID is proposed to be used with 64x64 CelebA and 64x64 ImageNet [Two time-scale update rule for training GANs]. So we think it is fine to use FID with 64x64 patches.
> * __Q3__. Sure, we can draw the perception-distortion trade-off as Fig. 1 of [Muckley 2023], and the result is shown in https://ibb.co/DrtyXTG. We choose this way because we can compare to other methods easily, while the way of [Muckley 2023] can only compare to ourselves as we can not match the bitrate of other methods without selecting the bitrate of our base model carefully.
> * __Q4__. Thanks for this advice, we will revise it into divergence-based perception to distinguish it with human perception.

---

> > ### Comment · Reviewer_VEis · 2023-11-20
> >
> > I would like the authors for replying to my review comments. I am satisfied with the answers. As I have already recommended to accept the paper, I maintain my original rating.

---

### Official Review · Reviewer_CP1y · 2023-11-02

**Soundness:** 3 good
**Presentation:** 4 excellent
**Contribution:** 4 excellent
**Rating:** 8
**Confidence:** 4

**Summary:**

This paper reveals the relationship between idempotence and perceptual image compression. There are two important theorems derived in this paper: (1) Perceptual quality (i.e., conditional generative model) brings idempotence. (2) Idempotence brings perceptual quality, with unconditional generative model. A specific idea in this paper is a new paradigm to achieve perceptual image compression by applying generative model inversion with pretrained models, which is conceptually simple but useful in practice. Experimental results also demonstrate the effectiveness of this idea  (Table 1 and Figure 3).

**Strengths:**

The relationship between idempotence and perceptual image compression revealed in this paper is surprising and insightful. The conclusion would be unintuitive before I successfully follow all the derivations in this paper. In fact, the derivations in this paper are not difficult. Theorems 1&2 well summarize the main conclusions of this paper. If there is no technical errors regarding the derivations (at least I didn't find), the proposed idea in this paper, i.e., using generative model inversion to improve perceptual image compression (Section4.1), may form a new paradigm in this field, which is sound in theory and useful in practice. The pratical implementation can be summarized as: first use pretrained generative models to produce an image $\hat{X}$ and encode it into $f_0(\hat{X})$, then constrain the idempotence between $f_0(\hat{X})$ and Y using gradient descent-based inversion. These steps are easy to implement with a pretrained codec and a pretrained unconditional generative model.

Experimental results clearly demonstrate the effectiveness of this idea. Also, the presentation of this paper is great. Overall, I would be happy to accept this paper although I have two major concerns as written in the weaknesses part.

**Weaknesses:**

Despite the above strengths of this paper, there are two main issues especially after I ran the code provided as supplementary material.
(1) The time complexity of generative model inversion should be considered in practice (Table 4), which I personally believe there is space to improve it in the future. For example, now the authors are using DDPM with 1000 steps. There are some latest unconditional generative models largely reducing the number of generation steps.
(2) The issue regarding the image resolution should be paid with more attention. We can imagine that the gradient descent-based inversion would be hard to be implemented on high-resolution images. In the experimental section of this paper, it is stated like "central crop image by their short edge and rescale them to 256 × 256", which means almost all experiments are performed on this scale. On the one hand, some baseline models are trained for high-resolution generation or generative compression, and it may be unfair to directly compare with their pretrained models. On the other hand, it implies that the current method is hard to be applied to high-resolution images without cropping.

**Questions:**

See the abovementioned weaknesses.
In addition, the approach proposed in this paper can utilize pre-trained unconditional generative model for compression at different bitrates. In Figure 9, it seems the advantage of the proposed method is more significant at 0.17bpp, compared with original ELIC. However, the result at 0.11 bpp seems to be weird, which is definitely what we do not want in practice. Is there is potential solution or idea to solve this issue?

---

> ### Author Response · Authors · 2023-11-13
> **Rebuttal by Authors**
>
> Thanks for your detailed review. And we are glad to provide our answer to your questions:
>
> * __W1__. As we have discussed in limitation, the speed is an issue shared by all diffusion-inversion approaches. Currently, the latest preprint in this area [Prompt-tuning latent diffusion models for inverse problems] using fast samplers, such as DDIM [Denoising Diffusion Implicit Models], still need the same 1000 step as DDPM. Even though DDIM works well with 50 steps for non-inverse problems. We have also attempted DDIM with fewer steps by ourselves, but we can not make it work. Thus, simply chaning DDPM to faster sampler does not work. The diffusion-inversion community is very active and we believe that the situation can be improved soon.
> * __W2__. As we have discussed in limitation, we agree that the resolution is a problem. We have done some very early attempt using diffusion model with larger resolution. For example, our approach works with 512x512 Stable Diffusion 1.5 and null text prompt. And a visual example with [Balle 2018] and ImageNet 512x512 Image is: https://ibb.co/xCd9gMB.
> * We have not tried 1024x1024 Stable Diffusion 2.0. Note that as Stable Diffusion is a latent diffusion, the latent space size is much smaller than pixel size. And thus its complexity does not grow that much. Currently our higher-resolution codec relies on higher-resolution diffusion model. We are also aware that there are other pathch-wise solutions [DiffCollage: Parallel Generation of Large Content with Diffusion Models], but they are not the mainstream in diffusion community and we have not given them a try.
> * For the baselines. we note that all the baselines are trained on 256x256 images. And to secure fair comparsion, we also provide the result of HiFiC and ILLM re-trained on 256x256 FFHQ and ImageNet in Tab.2.
> * __Q1__. The weird reconstruction is due to an improper hyper-parameter. More specifically, the scale parameter $\zeta=0.3$ is too small, which lead to the reconstruction deviate from original image. We have shown additional results varying $\zeta=0.6$ in https://ibb.co/XXRQN2F. Clearly use $\zeta=0.6$ improve the result.

---

> > ### Comment · Reviewer_CP1y · 2023-11-21
> >
> > Thanks for your responses. I will keep my score as 8. In addition, there is a concurrent paper (also a submission to ICLR2024): Idempotent Generative Models, which proposed a related idea but from a quite different view. Perhaps the authors may consider add a short discussion about this concurrent work in the final version.

---

> > > ### Author Response · Authors · 2023-11-21
> > >
> > > Thanks for the reminder. We have been awared of this submission by searching "Idempotent" as keyword. We are considering our relationship with it.

---

### Official Review · Reviewer_SHZQ · 2023-11-03

**Soundness:** 3 good
**Presentation:** 3 good
**Contribution:** 3 good
**Rating:** 8
**Confidence:** 4

**Summary:**

This paper studies the relation between the idempotence and perceptual image compression and find that the two are highly correlated. Specifically, the authors theoretically prove that: (1). conditional generative model-based perceptual codec satisfies idempotence; (2). Unconditional generative model with idempotence constraint is equivalent to conditional generative codec.
They also propose a new paragidm of perceptual image codec by inverting unconditional generative model with idempotence constraints, based on the above findings. The proposed method outperforms state-of-the-art methods in terms of perceptual quality metric, such as Frechet Inception Distance (FID).

**Strengths:**

-  This paper is the first to study the relation between the idempotence and perceptual image compression methods.
-  The findings are very interesting and motivating, the high correlation between the idempotence and perceptual image compression may benefit both above compression areas.
-  The theoretical analysis is rigorous and solid.
-  The proposed new compression method achieves better perceptual quality than the competing methods.

**Weaknesses:**

- I admit that the findings of high correlation between  the idempotence and perceptual image compression are intersting and meaningful. However, the  subsequent idea of "perceptual image compression by inversion" does not bring new insights or knowledge. Apply generative model inversion on the low-level tasks such as super-resolution has been well studied for several years. The proposed method seems to just change the task from super-resolution to image compression. Applying a proven methodology to a similar task is, in my opinion, hardly meets the ICLR's bar.

**Questions:**

- In the experiments, a MSE optimized codec is used as the base model. Have the authors tried other codecs (perceptual optimized or mixed)?
- Could the authors provide more results on other metrics, such as KID, LPIPS?

---

> ### Author Response · Authors · 2023-11-13
> **Rebuttal by Authors Part I**
>
> Thanks for your detailed review. And we are glad to provide our answer to your questions:
>
> * __W1__. First, migrating from super-resolution to codec is not that straightforward. Unlike super-resolution and other restortation, codec is highly non-linear. And many previously proposed approaches that work well for super-resolution, such as [Menon 2020] [Daras 2021] [Chung 2022a], fail. As we have shown in Tab. 1, a thorough curation of different generative model and inversion approach is required to make it actually work. Furthermore, in Sec. 4.2, we additionally propose to use the straight through estimation to make codec differentiable, and the x-domain y-domain constraint. Those techniques are codec specific and have never been studied in super-resolution.
> * Second, indeed we borrow the empirical approach from super-resolution. However, we also give back theoretical understanding. Despite inversion-based super-resolution has been studied for years empirically, their theoretical relationship with conditional model based super-resolution, and distortion-perception trade-off is in general unknown. PULSE [Menon 2020] is the poineer of this area, and they justify their approach by "natural image manifold" assumption. And later works in inversion-based super-resolution follow PULSE's story. On the other hand, our __Theorem 1__, __Theorem 2__ only limit the operator $f(.)$ to be deterministic, and it need not to be a codec. Then, our __Theorem 1__, __Theorem 2__ and __Corollary 1__ are also appliable to inversion based super-resolution, at least when the degeneration operator is deterministic. In other words, our theoretical results also implies that:
>     * __Theorem 1'__ (super-resolution version) Conditional generative super-resolution also satisfies idempotence, that is, the up-sampled image can downsample into low-resolution image.
>     * __Theorem 2'__ (super-resolution version) Inversion-based super-resolution is theoretically equvalient to conditional generative super-resolution.
>     * __Corollary 1'__ (super-resolution version) Inversion-based super-resolution satisfies the theoretical results of distortion-perception trade-off, that is the MSE is at most double of best MSE.
> * We believe that our result provides non-trivial theoretical insights to inversion-based super-resolution community. For example, [Menon 2020] claim that the advantage of inversion-based super-resolution over the conditional generative super-resolution is that inversion-based super-resolution "downscale correctly". However, with our theoretical result, we know that ideal conditional generative super-resolution also "downscale correctly". Currently they fail to achieve this due to implemention issue. Another example is that most inversion-based super-resolution [Menon 2020] [Daras 2021] [Chung 2022a] report no MSE comparsion with sota MSE super-resolution, as it is for sure that their MSE is worse and there seems to be no relationship between their MSE and sota MSE. However, with our theoretical result, we know that their MSE should be smaller than 2x sota MSE. And they should examine whether their MSE falls below 2x sota MSE.

---

> ### Author Response · Authors · 2023-11-13
> **Rebuttal by Authors Part II**
>
> * __Q1__. By [Blau 2018] and __Corollary 1__, the MSE of inversion-based perceptual codec is bounded by 2x base codec. Using a perceptual codec as base codec:
>     * does not help with perceptual quality, as theoretically any base codec inversion achieves perfect perceptual quality.
>     * does loosen MSE upperbound, as the current upperbound is 2x MSE of a perceptual codec. And it is very likely that the MSE of our codec grows larger.
> * Therefore, we choose to use a MSE optimized codec, to obtain the tightest MSE bound possible.
> * __Q2__. Sure, we have evaluated our approach additionally on KID and LPIPS:
>
>     | Method              | FFHQ      |          | ImageNet  |          | COCO      |          | CLIC      |          |
>     |---------------------|-----------|----------|-----------|----------|-----------|----------|-----------|----------|
>     |                     | BD-logKID↓| BD-LPIPS↓| BD-logKID↓| BD-LPIPS↓| BD-logKID↓| BD-LPIPS↓| BD-logKID↓| BD-LPIPS↓|
>     | _MSE Baselines_     |           |          |           |          |           |          |           |          |
>     | Hyper               | 0.0000    | 0.0000   | 0.0000    | 0.0000   | 0.0000    | 0.0000   | 0.0000    | 0.0000   |
>     | ELIC                | -0.2320   | -0.0402  | -0.348    | -0.0580  | -0.236    | -0.0623  | -0.4067   | -0.0595  |
>     | BPG                 | 0.1506    | -0.0102  | 0.0274    | -0.0102  | 0.1267    | -0.0116  | 0.03985   | -0.00795 |
>     | VTM                 | -0.2320   | -0.0311  | -0.2975   | -0.0476  | -0.2157   | -0.0504  | -2.049    | -0.04812 |
>     | _Conditional-based_ |           |          |           |          |           |          |           |          |
>     | HiFiC               | -3.132    | -0.1087  | -2.274    | -0.1724  | -2.049    | -0.1723  | -1.925    | -0.1483  |
>     | HiFiC*              | -4.261    | -0.1102  | -2.780    | -0.1734  | N/A       | N/A      | N/A       | N/A      |
>     | Po-ELIC             | -3.504    | -0.1047  | -2.877    | -0.1674  | -2.671    | -0.1687  | -2.609    | -0.145   |
>     | CDC                 | -2.072    | -0.0600  | -1.968    | -0.0985  | -1.978    | -0.1011  | -2.122    | -0.08437 |
>     | ILLM                | -3.418    | -0.1092  | -2.681    | -0.1809  | -2.62     | -0.1805  | -2.882    | -0.1547  |
>     | ILLM*               | -4.256    | -0.1062  | -2.673    | -0.1784  | N/A       | N/A      | N/A       | N/A      |
>     |_Unconditional-based_|           |          |           |          |           |          |           |          |
>     | Proposed (Hyper)    | -5.107    | -0.0859  | -4.271    | -0.0576  | -4.519    | -0.0826  | -3.787    | -0.0557  |
>     | Proposed (ELIC)     | -5.471    | -0.0987  | -5.694    | -0.1062  | -5.36     | -0.113   | -4.046    | -0.0796  |
> * The result of KID has same trend as FID, which means that our approach is sota. While many other approaches outperform our approach in LPIPS. This result is expected, as:
>     * KID is a divergence based metric. And our approach is optimized for divergence.
>     * LPIPS is a image-to-image distortion. By distortion-perception trade-off [Blau 2018], perception optimized approaches can not achieve SOTA LPIPS.
>     * All other perceptual codec (HiFiC, ILLM, CDC, Po-ELIC) except for ours use LPIPS as loss function during training.
>     * ILLM [Muckley 2023] also achieves sota FID, KID and visual quality at that time, but its LPIPS is outperformed by a autoencoder trained with LPIPS.

---

> > ### Comment · Reviewer_SHZQ · 2023-11-22
> >
> > Thanks for the detailed reply and the new experiment results. My major concerns are well addressed. I will change my rating to 8.

---

### Author Response · Authors · 2023-11-15
**Summary of Revision**

Thanks for your detailed review. We have uploaded the revised main text and appendix, with all the revisions marked in blue. Due to the space limitation, we have to put many of amendment in appendix. Below is a summary of revisions:
* We emphasis that we use the word "perception" for [Blau 2018]'s divergence-based perception and we use the word "human perception" for real human's perception (as suggested by VEis).
* We include the discussion on why MSE-optimized codec is used instead of perceptual codec as base for inversion (as suggested by SHZQ).
* We include an example to better understand __Theorem 1__, __Theorem 2__ (as suggested by 7mRQ).
* We add additional explaination on how our theoretical result might impact inversion-based super-resolution community (as suggested by SHZQ).
* We include the discussion on why some inversion approaches fail (as suggested by CP1y).
* We include the discussion on the fail case of Fig. 9 (0.11) bpp and how to fix it (as suggested by CP1y).
* We include additional results on KID and LPIPS (as suggested by SHZQ, VEis and 7mRQ).
* We include additional results on the difference of Torch HifiC and Tensorflow HiFiC (as suggested by VEis).
* We include an alternative way to present perception-distortion trade-off (as suggested by VEis).

We hope that we have addressed your concerns and we are glad to provide additional clarifications.

---

### Meta-Review · Area_Chair_Fc4e · 2023-12-11

**Metareview:**

This paper introduces a new paradigm of perceptual image codec by inverting unconditional generative model with idempotence constraints. All reviewers appreciate the findings of the paper and recommend acceptance.

**Justification For Why Not Higher Score:**

The interest to the broader community seems a bit niche.

**Justification For Why Not Lower Score:**

This paper introduces a new paradigm of perceptual image codec. The findings of the relation between the idempotence and perceptual image compression methods is interesting.

---

### Decision · Program_Chairs · 2024-01-16

Accept (spotlight)